# Tethering of an E3 ligase by PCM1 regulates the abundance of centrosomal KIAA0586/Talpid3 and promotes ciliogenesis

Lei Wang[†], Kwanwoo Lee[†], Ryan Malonis, Irma Sanchez, Brian D Dynlacht*

Department of Pathology, New York University Cancer Institute, New York University School of Medicine, New York, United States

**Abstract** To elucidate the role of centriolar satellites in ciliogenesis, we deleted the gene encoding the PCM1 protein, an integral component of satellites. *PCM1* null human cells show marked defects in ciliogenesis, precipitated by the loss of specific proteins from satellites and their relocation to centrioles. We find that an amino-terminal domain of PCM1 can restore ciliogenesis and satellite localization of certain proteins, but not others, pinpointing unique roles for PCM1 and a group of satellite proteins in cilium assembly. Remarkably, we find that PCM1 is essential for tethering the E3 ligase, Mindbomb1 (Mib1), to satellites. In the absence of *PCM1*, Mib1 destabilizes Talpid3 through poly-ubiquitylation and suppresses cilium assembly. Loss of PCM1 blocks ciliogenesis by abrogating recruitment of ciliary vesicles associated with the Talpid3-binding protein, Rab8, which can be reversed by inactivating Mib1. Thus, PCM1 promotes ciliogenesis by tethering a key E3 ligase to satellites and restricting it from centrioles.

*For correspondence: Brian.
Dynlacht@nyumc.org

[†]These authors contributed
equally to this work

**Competing interests:** The
authors declare that no
competing interests exist.

**Reviewing editor:** Mark Winey,
University of Colorado Boulder,
United States

## Introduction

Primary cilia are antenna-like organelles that protrude from the surface of many differentiated cells and serve to orchestrate key signaling events required for development (reviewed in *Kobayashi and Dynlacht, 2011*). The cilium and centrosome comprise several hundred polypeptides, although the functions of many of these proteins remain uncharacterized. Recent efforts, however, have begun to shed considerable light on processes and constituents that play a key role in the biogenesis, structure, and function of these organelles.

Cilium assembly is initiated in cultured cells that have exited the cell cycle after mitogen deprivation. The process commences with the docking of ciliary vesicles at the distal appendages of the basal body, proceeds with the elongation of an axoneme concomitant with recruitment of Rab8-associated vesicles, and culminates with the fusion of the nascent cilium with the plasma membrane (*Kobayashi and Dynlacht, 2011*; *Sorokin, 1962*). Other proteins, including the BBSome and Intraflagellar Transport (IFT) complexes, contribute to growth, assembly, and maintenance of the cilium (*Ishikawa and Marshall, 2011*). An extensive array of centriolar and basal body proteins has also been shown to play a critical role in ciliogenesis. Among these key regulators, Talpid3 is essential for ciliogenesis, and mutations in the gene encoding this protein lead to a panoply of developmental defects in model organisms and human ciliopathies, including Joubert Syndrome (*Alby et al., 2015*; *Bachmann-Gagescu et al., 2015*; *Bangs et al., 2011*; *Davey et al., 2006*; *2007*; *Malicdan et al., 2015*; *Roosing et al., 2015*; *Stephen et al., 2015*). We have shown that Talpid3 associates with Rab8, a component of ciliary vesicles, and silencing Talpid3 compromises the efficient recruitment of ciliary vesicles to basal bodies (*Kobayashi et al., 2014*).

Other proteins, including Mindbomb 1 (Mib1), act to antagonize ciliogenesis (*Villumsen et al., 2013*). Mib1 was first reported as an E3 ligase associated with Notch signaling (*Itoh et al., 2003*), although connections, if any, between a role for Mib1 in Notch signaling and ciliogenesis remain unknown. More recently, Mib1 was shown to associate with centriolar satellites (*Villumsen et al., 2013*), electron-dense, cytoplasmic granules defined by the presence of PCM1. Centriolar satellites (CS) are thought to play a role in trafficking of cargoes to the centrosome and cilium and are thought to promote ciliogenesis. Recently, an inhibitory role for centriolar satellite proteins has also been proposed, although many questions remain regarding the assembly of these granules and their function in ciliogenesis (reviewed in *Tollenaere et al., 2015*). For example, Mib1 was shown to associate with AZI1/Cep131 (hereafter Cep131) and PCM1 at centriolar satellites (*Villumsen et al., 2013*), and while Mib1 over-expression was shown to promote ubiquitylation of both proteins, no impact on their stability was observed. Thus, although Mib1 over-expression inhibits ciliogenesis, the mechanistic connections, if any, between Cep131 and PCM1 ubiquitylation and the inhibition of ciliogenesis were unclear. In other studies, centriolar satellites were shown to play a second, restrictive role in ciliogenesis by regulating the timely redistribution of the BBSome subunit, BBS4 (*Stowe et al., 2012*). Thus, centriolar satellites could play opposing roles by promoting or antagonizing initiation of the ciliogenesis program.

Here, we find that PCM1 plays a prominent role in ciliogenesis by sequestering proteins in the centriolar satellite compartment. Importantly, PCM1 restricts Mib1 to satellites, preventing its translocation to centrioles, which would otherwise promote Talpid3 destabilization, a failure to recruit Rab8a, and inhibition of cilium assembly. Moreover, using PCM1 knock-out cells, we show that the ability of this protein to promote ciliogenesis by partitioning satellite proteins is not universal, allowing us to identify distinct roles in cilium assembly. Our studies provide novel insights into mechanistic roles for PCM1 in ciliogenesis and explain how centriolar satellites can play a positive role in ciliogenesis by antagonizing inhibitors of this process.

## Results

### *PCM1* knock-out cells display a constellation of defects and cannot ciliate

In an effort to study the role of centriolar satellites in the assembly of primary cilia, we used CRISPR/Cas9-mediated gene-editing in retinal pigment epithelial cells (RPE1) to ablate *PCM1*, a central CS component. We isolated a clone with frame-shift mutations in exon 2 of *PCM1*, and the cell line was devoid of PCM1 protein (*Figure 1C,D*, and *Figure 1—figure supplement 1*). We examined this cell line by immunofluorescence and transmission electron microscopy (TEM) and found that in contrast with previous studies, in which ciliation was reduced by 20–50% after knock-down, gene ablation resulted in a complete failure to ciliate when cells were made quiescent through serum deprivation (*Figure 1A*, *Figure 1—figure supplement 1*, and data not shown)(*Lopes et al., 2011*; *Stowe et al., 2012*). To rule out off-target effects, we showed that expression of Myc-tagged mouse or EYFP-tagged human PCM1 in these cells largely rescued the ciliogenesis defect (*Figure 1A and C* and *Figure 2—figure supplement 2*). In subsequent experiments, we primarily used mouse PCM1, since its expression was more stable and provided near-equivalent levels across our deletions series (*Figure 2B*).

Next, we examined a cohort of proteins known to associate with centrioles, centriolar satellites, and primary cilia in *PCM1* null cells by western blotting and immunofluorescence. We found that *PCM1* knock-out cells completely lacked centriolar satellites, which were visualized through immunofluorescent detection with a panel of antibodies against known satellite proteins (Cep131, Cep290, Cep90, and Mib1), and this defect could be rescued by expression of Myc-PCM1 (*Figure 1—figure supplement 1* and *Figure 3*). Interestingly, the disappearance of each of these proteins at satellites was accompanied by their appearance at centrioles (*Figure 1—figure supplement 1*). Consistent with these observations, the satellite proteins Cep72, Cep90, and Cep290 were also retained at centrosomes upon PCM1 knock-down (*Kim et al., 2012*; *Lopes et al., 2011*; *Stowe et al., 2012*). Moreover, we observed an overall reduction in abundance of several satellite proteins, including Cep131, BBS4, and Cep90 (*Figure 1D*). On the other hand, Mib1 levels were elevated upon loss of PCM1, and other proteins that populate the satellite compartment, centrioles, or ciliary vesicles, such as

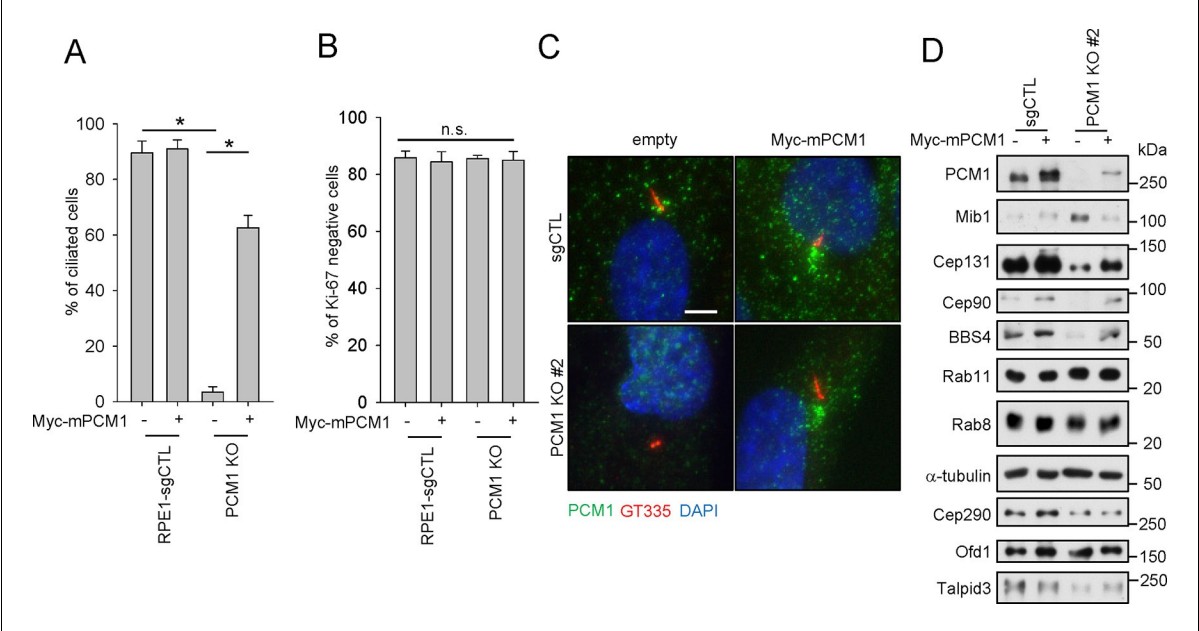

**Figure 1.** PCM1 regulates the abundance of centriolar satellite proteins and is essential for ciliogenesis. CRISPR/Cas9 deletion of PCM1 inhibits ciliogenesis. (A) and (B) Control and PCM1 KO RPE1 cells were infected with empty or Myc-mPCM1 lentivirus for 72 hr and then serum starved for 48 hr. The cells were immuno-stained with GT335 and Ki-67 antibodies. (A) Ciliated cells and (B) Ki-67-negative cells were counted; n ≥ 100 per sample in three independent experiments. Error bars, SEM. *p<0.05 (C) Ectopic PCM1 can rescue the ciliogenesis defect of *PCM1* KO RPE cells. Control and *PCM1* KO RPE1 cells were infected with empty or Myc-mPCM1 lentivirus for 72 hr, serum starved for 48 h, and immuno-stained with antibodies against glutamylated tubulin (GT335, red), and PCM1 (green) and with DAPI (blue). Representative images are shown. Scale bar, 5 µm. (D) Altered levels of centriolar satellite proteins in *PCM1* KO RPE1 cells. Control and *PCM1* KO RPE1 cells were infected with control or Myc-mPCM1 lentivirus for 72 hr and subjected to western blot analysis with the indicated antibodies.

The following figure supplement is available for figure 1:

**Figure supplement 1.** PCM1 gene-editing and the localization of Cep290, Cep131 and Cep90 in *PCM1* KO cells.

Cep290, Ofd1, Rab8, and Rab11, were unaffected, suggesting that the complete loss of PCM1 has specific effects on distinct centrosomal and peri-centrosomal components. We ruled out the possibility that these alterations resulted from changes in transcript levels or stability, rather than protein stability, by assessing steady state levels through quantitative reverse transcriptase-coupled PCR (qRT-PCR)(data not shown).

Our studies thus establish that PCM1 is absolutely required for (1) assembly of centriolar satellites and retention of proteins in this compartment and (2) maintenance of appropriate levels of proteins required to regulate ciliogenesis. In the absence of PCM1, a specific group of proteins known to localize to centrioles and centriolar satellites partitions exclusively to centrioles.

## PCM1 selectively interacts with proteins to promote centriolar satellite organization and ciliogenesis

To elucidate the domains of PCM1 required to restore primary cilia, we performed rescue experiments by expressing full-length mouse PCM1, or several truncations thereof, in *PCM1* null cells. Among this group of mutants, we found that only fragments spanning residues 1–1200 and 1–1500 PCM1 were able to rescue ciliogenesis in a manner comparable to the full-length protein (*Figure 2A*). In addition, only these two fragments were capable of forming PCM1-positive foci in the cytoplasm, and both exhibited a centriolar satellite-like localization similar to that observed with full-length PCM1 (*Figure 3* and data not shown). Consistent with these observations, both fragments were able to interact with several satellite proteins, including Mib1, Cep290, Cep131, Cep72, and OFD1 (*Figure 2B* and data not shown). Furthermore, PCM1 1–1200 rescued the normal localization of Mib1 and Cep131 at centriolar satellite-like PCM1 foci and prevented aberrant localization to

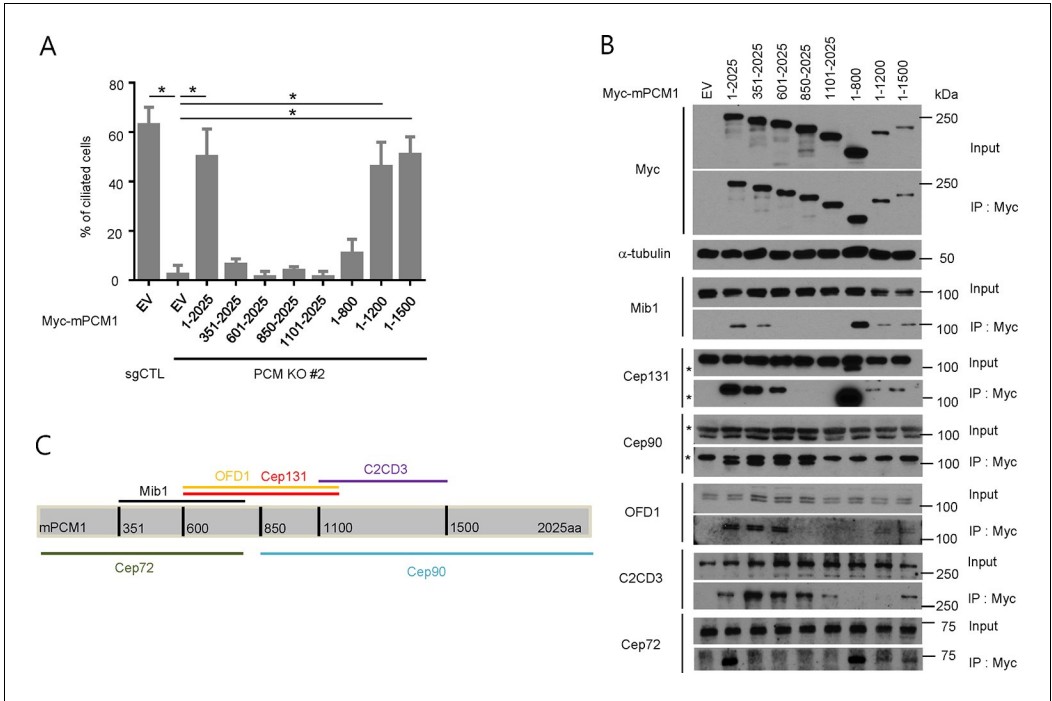

**Figure 2.** PCM1 promotes centriolar satellite organization and ciliogenesis through an amino-terminal domain. (**A**) Mapping of PCM1 domains required to rescue the ciliogenesis defect in *PCM1* null cells. Control and *PCM1* KO RPE1 cells were infected with virus containing the empty vector (EV) or the indicated Myc-mPCM1 fragments. Cells were infected with lentivirus for 3 days and serum starved for 48 hr. The cells were immuno-stained with antibodies against IFT88, detyrosinated tubulin, and the Myc epitope. Ciliated cells were counted within the Myc-positive population. n ≥100 were counted per sample in two independent experiments. Error bars, SD. *p<0.05. (**B, C**) Identifying PCM1 domains required to interact with centriolar satellite proteins. (**B**) HEK293T cells were transfected with plasmids corresponding to the empty vector (EV) or the indicated Myc-mPCM1 fragments for 48 h, and lysates subjected to immunoprecipitation with anti-Myc antibody. The inputs and the immunoprecipitates were analyzed by western blotting with the indicated antibodies. Asterisks indicate non-specific cross-reactive bands. We surmise that the band detected by anti-Cep131 antibodies in cell extracts expressing fragment 1–800 is cross-reactive, as this species consistently migrates at a position distinct from Cep131. (**C**) Schematic representation of protein-protein interacting domains in PCM1 based on immunoprecipitation data in (**B**).

The following figure supplements are available for figure 2:

**Figure supplement 1.** Centrosomal localization of Mib1 in PCM1 knock-down cells.

**Figure supplement 2.** PCM1 promotes ciliogenesis through an amino-terminal domain.

centrioles seen in PCM1 KO cells infected with the control virus (*Figure 3*), confirming that this fragment is sufficient to recruit centriolar satellite proteins and prevent their aberrant centriolar localization.

To further understand how PCM1 promotes ciliogenesis, we mapped domains in PCM1 responsible for interaction with other centriolar satellite components involved in ciliogenesis (reviewed in *Tollenaere et al., 2015*). Consistent with the notion that PCM1 serves as a platform to assemble centriolar satellites, we found that PCM1 interacted with Cep90 and C2CD3 through domains that overlapped with Cep131- and OFD1-binding regions but that were distinct from the CEP72- and Mib1-interacting domain. In line with previous reports (*Kamiya et al., 2008*; *Lopes et al., 2011*), PCM1 could associate with OFD1 through residues 600–1200 (*Figure 2B,C*). We also showed, for the first time, that PCM1 interacts with Mib1, Cep72, Cep90, C2CD3, and Cep131 through distinct, but sometimes overlapping, binding domains. Cep290 interacted with multiple PCM1 fragments, preventing us from more precisely mapping interacting regions (data not shown). Most strikingly, the PCM1 fragment spanning residues 1–1200, which could restore cilia and centriolar satellites, interacted with Mib1, OFD1, Cep131 and Cep290, but not Cep90 or C2CD3. In contrast, the PCM1 fragment containing residues 601–2025, which failed to restore cilia or centriolar satellites,

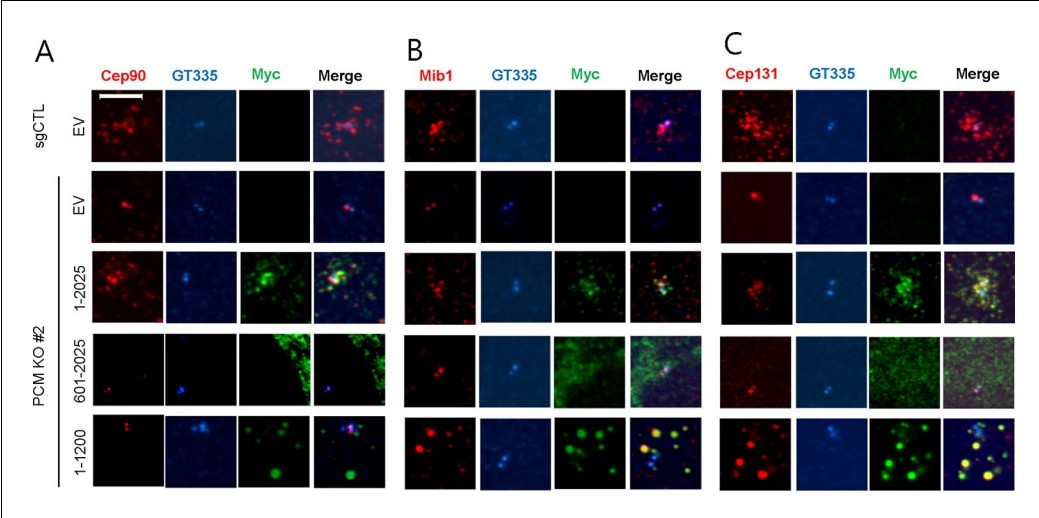

**Figure 3.** The amino-terminal domain of PCM1 recruits specific centriolar satellite proteins. Control and PCM1 KO RPE1 cells were infected with lentivirus containing empty vector or the indicated Myc-mPCM1 fragments for 3days and stained with the antibodies against Myc, GT335, Cep90, Mib1, and Cep131. Diffuse cytoplasmic staining of PCM1 was observed with the fragment spanning residues 601–2025. Scale bar, 5 µm.

The following figure supplement is available for figure 3:

**Figure supplement 1.** The amino-terminal domain of PCM1 recruits specific centriolar satellite proteins.

interacted strongly with most of the proteins we analyzed, except for Mib1 and Cep72 (*Figure 2B*). These results indicate a potentially important role for PCM1-Mib1 and PCM1-Cep72 interactions in both ciliogenesis and satellite assembly, while suggesting that other PCM1 interactions are not sufficient to restore these processes.

Next, we asked whether the interactions identified above are required for the centriolar satellite localization of each of these factors. We re-introduced full length mouse PCM1, or several truncations, in *PCM1* null cells and visualized centriolar satellite staining for Mib1, Cep90, and Cep131. Consistent with our mapping data, the amino-terminal 1200 residues of PCM1 rescued the centriolar satellite localization of Mib1 and Cep131, but not Cep90 (*Figure 3* and *Figure 3—figure supplement 1*). Further, the rescue of Cep131 centriolar satellite localization by the amino-terminal 1200 residue fragment of PCM1 (*Figure 3C*) could be explained by binding to PCM1, through a PCM1-Mib1-Cep131 interaction (*Villumsen et al., 2013*), or a combination of interactions.

These results lead to several important conclusions. First, PCM1 promotes ciliogenesis, most likely through interactions with a specific cohort of proteins, including Mib1, OFD1, Cep131, Cep72, Cep290, and/or other proteins that we have not tested here, whereas interactions with other satellite proteins, namely, Cep90 and C2CD3 do not appear to be required for ciliogenesis. Our results also suggest that proper centriolar satellite localization of certain pro-ciliogenic proteins, such as Cep90, is not universally required for ciliogenesis.

## PCM1 anchors Mib1 at centriolar satellites

Next, we investigated the potential interplay between PCM1 and Mib1 because Mib1 levels were elevated in *PCM1* knock-out cells, in striking contrast with the behavior of other centriolar satellite proteins that we examined (*Figure 1D*), and because the domain in PCM1 essential for ciliogenesis can interact with Mib1 (*Figure 2*). First, we examined the localization of Mib1 in wild-type and *PCM1* null cells. To help visualize centrosomes more accurately without the confounding effects of centriolar satellites, within which they are interspersed, we also treated cells with nocodazole, which effectively eliminates the latter, microtubule-dependent structures. Consistent with prior studies (*Villumsen et al., 2013*), Mib1 did not co-localize with the centriolar marker, centrin, but primarily localized to satellites in untreated control cells, and this observation was made more evident by treatment of these cells with nocodazole (*Figure 4A*). Interestingly, we found that in *PCM1* knock-

out cells, the loss of satellites, observed in the presence or absence of nocodazole, was accompanied by re-localization of Mib1 to centrioles (*Figure 4A–D*). Furthermore, we observed identical results when PCM1 was ablated with an siRNA (*Figure 2—figure supplement 1*). These results suggest that depletion of PCM1 triggers the destabilization of satellites and thereby promotes the re-localization and accumulation of Mib1 protein at centrioles. Indeed, trafficking to centrioles might stabilize Mib1 at this location (*Figure 1D*). PCM1 could therefore function to actively sequester Mib1 at satellites. To test this possibility, we rescued *PCM1* null cells with Myc-PCM1 and examined centriolar staining in nocodazole-treated cells. We found that re-expression of PCM1 restored both the exclusion of Mib1 from centrosomes and native levels of this protein (*Figures 1D* and *4C,D*). Re-expression of the amino-terminal PCM1 1–1200 fragment also restored the exclusion of Mib1 from centrosomes and reduced Mib1 protein levels (*Figure 3B* and *Figure 3—figure supplement 1*), consistent with an ability of PCM1 to actively sequester Mib1. Moreover, ectopic expression of Mib1 resulted in its re-localization to centrioles (see *Figure 7G,H*), suggesting that the number of binding sites for Mib1 (within centriolar satellites) may be saturable.

Taken together with our immuno-blot analysis, our studies suggest that PCM1 modulates Mib1 abundance and sequestration, and when PCM1 is limiting or lost and centriolar satellite integrity compromised, Mib1 is able to accumulate at centrioles.

## PCM1 regulates ciliogenesis through sequestration of Mib1

One study, published while our work was in progress, suggested that Mib1 suppresses ciliogenesis in growing cells (*Villumsen et al., 2013*), and we have confirmed that cilium assembly is dramatically enhanced upon its depletion in growing cells and abrogated in quiescent cells after its ectopic expression (*Figure 5—figure supplement 1*). Moreover, consistent with this observation, expression of a catalytically inactive, dominant-negative form of this E3 ligase (C985S) did not suppress cilium assembly (*Figure 5—figure supplement 1*). These findings suggested that the failure of serum-starved *PCM1* null cells to ciliate could stem from the enhanced expression and accumulation of Mib1 at centrioles (*Figures 1D* and *4A*). To directly test this hypothesis, we ablated Mib1 from these cells using two distinct Mib1 siRNAs (*Figure 5A*). Strikingly, we found that the removal of Mib1 partially restored the ability of serum-starved *PCM1* null cells to ciliate (*Figure 5B* and *Figure 5—figure supplement 1*). Likewise, expression of catalytically inactive Mib1 in *PCM1* KO cells partially rescued cilium assembly (*Figure 5C*), confirming that Mib1 E3 ligase activity can suppress cilium assembly if it is not tethered to the centriolar satellite compartment (*Figure 4A–D*).

Next, we asked whether the loss of Mib1 could revert ciliation defects in cell lines without PCM1 defects. To this end, we performed analogous experiments in *TALPID3* null RPE1 cells, which we generated using CRISPR/Cas9-mediated gene-editing (*Figure 5—figure supplement 2*). Like the *PCM1* knock-out, *TALPID3* null cells were unable to ciliate, but unlike our findings with *PCM1* null cells, *TALPID3* loss had no impact on Mib1 localization and was not rescued by Mib1 depletion (*Figure 5—figure supplement 2* and *Figure 5B*). These results suggest that PCM1, and by extension, centriolar satellites, play an important role in promoting primary ciliogenesis by sequestering and functionally antagonizing the Mib1 E3 ligase.

The loss of PCM1 led to the near-disappearance of several centriolar satellite proteins, namely, Cep131, Cep90, and BBS4 (*Figures 1D* and *5D*). We speculated that if PCM1 anchoring plays a critical role in restricting the localization and levels of Mib1 and other centriolar satellite proteins, then *PCM1* ablation could mimic the effects of ectopically expressing Mib1, which likewise suppresses cilium assembly. Consistent with this idea, ectopic expression of wild-type, but not catalytically inactive, Mib1 prompted the disappearance of PCM1, Cep131, and, to a lesser extent, Talpid3, in RPE1 cells (*Figure 5D*). This result was not restricted to RPE1 cells, since we observed similar or even more dramatic down-regulation of each of these proteins, particularly Talpid3, in U2OS cells (*Figure 5—figure supplement 3*). The impact of Mib1 was specific, since ectopic expression of Mib1 led to diminished levels of certain centriolar satellite proteins (PCM1, Cep131, Cep290, Cep90), but not others (Ofd1, BBS4) (*Figure 5—figure supplement 3*). Moreover, the abundance of centriolar/centrosomal proteins (Sas6) or modified tubulins (acetylated tubulin) was not affected (*Figure 5—figure supplement 3*). *PCM1* null cells have dramatically reduced overall levels of Cep131, which also localizes to centrioles (*Figure 1—figure supplement 1*), and expression of wild-type, but not inactive, Mib1 completely eliminated centrosomal Cep131 (*Figures 5D* and *Figure 5—figure supplement 3*). Expression of Mib1 led to the near-complete dissolution of centriolar satellites typified by PCM1 and

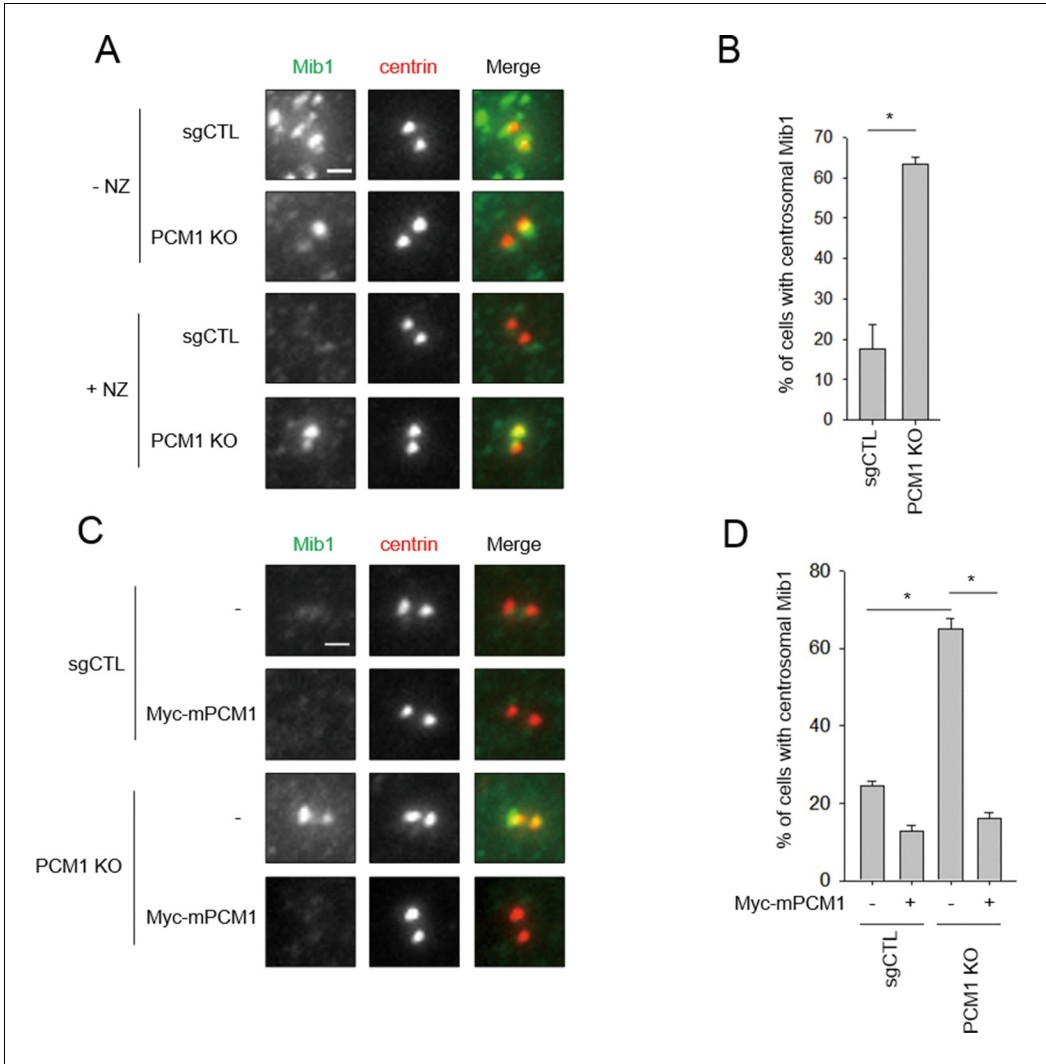

**Figure 4.** Mib1 re-locates to centrosomes in *PCM1* null cells. (**A**) Control and *PCM1* KO RPE1 cell were incubated with DMSO or nocodazole (NZ; 1μg/ml) for 8 hr and then immuno-stained with Mib1 (green) and centrin (red) antibodies (**A**) and (**B**). Representative images of centrosomes are shown (**A**). Scale bar, 1 μm. (**B**) Quantitation of nocodazole-treated cells in panel A. Cells with centrosomal Mib1 signal were counted in the NZ-treated group. n ≥ 100 per sample in two independent experiments. Error bars, SD. *p<0.05. (**C**) Ectopic PCM1 expression in PCM1 KO RPE cells prevents centrosomal localization of Mib1. Control and *PCM1* KO RPE1 cells were infected with empty or Myc-mPCM1 lentiviruses for 72 hr and incubated with NZ (1 μg/ml) for 8 hr (**C**) and (**D**). (**C**) Cells with centrosomal Mib1 signal were analyzed by immuno-staining with Mib1 (green) and centrin (red) antibodies. Representative images of centrosome are shown. Scale bar, 1 μm. (**D**) Quantitation of data in panel C. *N* ≥ 100 per sample in two independent experiments. Error bars, SD. *p<0.05.

Cep290 enrichment, whereas inactive Mib1 dramatically led to their stabilization (*Figure 5—figure supplement 3*). Consistent with this observation, depletion of Mib1 with two distinct siRNAs led to dramatic up-regulation of centriolar satellites, concomitant with Cep131, Cep290, and PCM1 stabilization (*Figure 5—figure supplement 3*). These results strongly suggest that Mib1 is able to selectively regulate the stability of multiple centriolar satellite and centrosomal proteins.

## Mib1 promotes the poly-ubiquitylation of Talpid3, Cep131, and PCM1

We sought to understand the mechanisms by which Mib1 could regulate the levels of centriolar satellite proteins, as well as centriolar proteins such as Talpid3, since it is required for ciliogenesis. In our previous proteomic screen for Talpid3-interacting proteins (*Kobayashi et al., 2014*), we identified Mib1 as a robust interactor. We confirmed this interaction by expressing either Mib1 or Talpid3 in cells and performing reciprocal immunoprecipitations (*Figure 6A*). We did not observe an

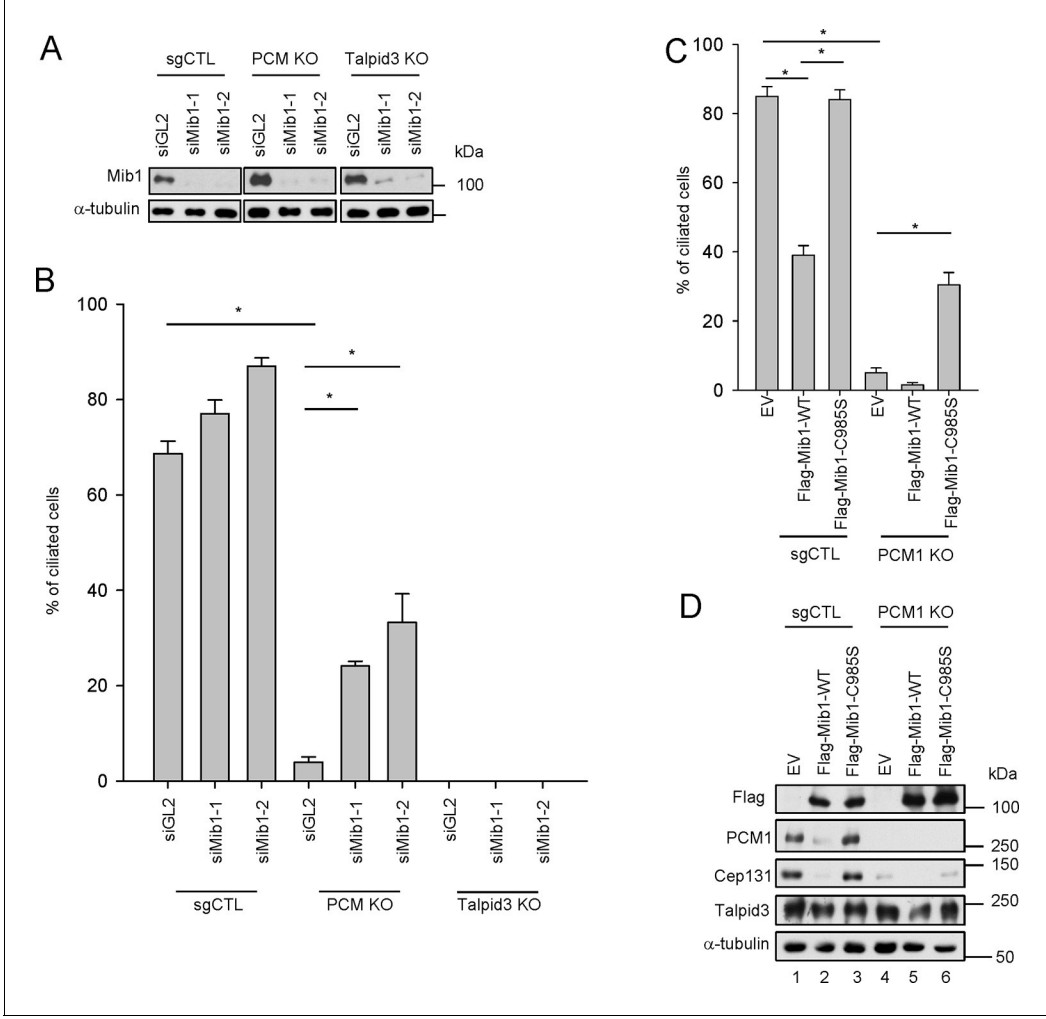

**Figure 5.** Mib1 is downstream of PCM1. (**A**) Validation of siRNA knockdown of Mib1. Control (*sgCTL), PCM1,* and *TALPID3* KO RPE1 cells were transfected with the indicated siRNAs corresponding to non-specific control (siGL2) or Mib1 (*siMIB1-1* or *siMIB1-2*), and lysates were subjected to western blot analysis with the indicated antibodies. (**B**) Mib1 depletion partially rescues the ciliogenesis defect in *PCM1* KO cells. Cell lines in panel A were transfected with siRNAs corresponding to non-specific control (*siGL2*) or Mib1 (*siMIB1-1* or *siMIB1-2*) for 48 hr and serum starved for 48 hr. Ciliated cells (n ≥ 100 per sample in three independent experiments) were analyzed by immuno-staining with GT335. Error bars, SEM. *p<0.05. (**C**) E3 ligase activity of Mib1 underlies ciliogenesis defect in *PCM1* KO cells. Control and PCM1 KO RPE1 cells were infected with empty (EV), Flag-Mib1-WT or Flag-Mib1-C985S lentiviruses for 72 hr and serum starved for 48 hr. Ciliated cells were analyzed by immuno-staining with GT335; n≥100 per sample were analyzed in two independent experiments. Error bars, SD. *p<0.05. (**D**) Control, *PCM1*, and *Talpid3* KO RPE1 cells were transfected with siGL2, *siMIB1-2*, or *siMIB1-3* for 48 hr and subjected to western blot analysis with the indicated antibodies.

The following figure supplements are available for figure 5:

**Figure supplement 1.** Mib1 suppress ciliogenesis.

**Figure supplement 2.** Generation of *TALPID3* knock-out cell lines.

**Figure supplement 3.** Mib1 regulates the stability of multiple centriolar satellite proteins.

interaction between endogenous Mib1 and Talpid3, since these proteins normally reside in discrete compartments; instead, we found that ectopic Mib1 could bypass this restriction. We generated a series of Talpid3 truncation mutants and showed that an amino-terminal fragment spanning residues 1–1000 interacted most strongly with Mib1, although individual fragments spanning either half of this fragment also interacted with Mib1, albeit much more weakly (*Figure 6B*). Next, we assessed

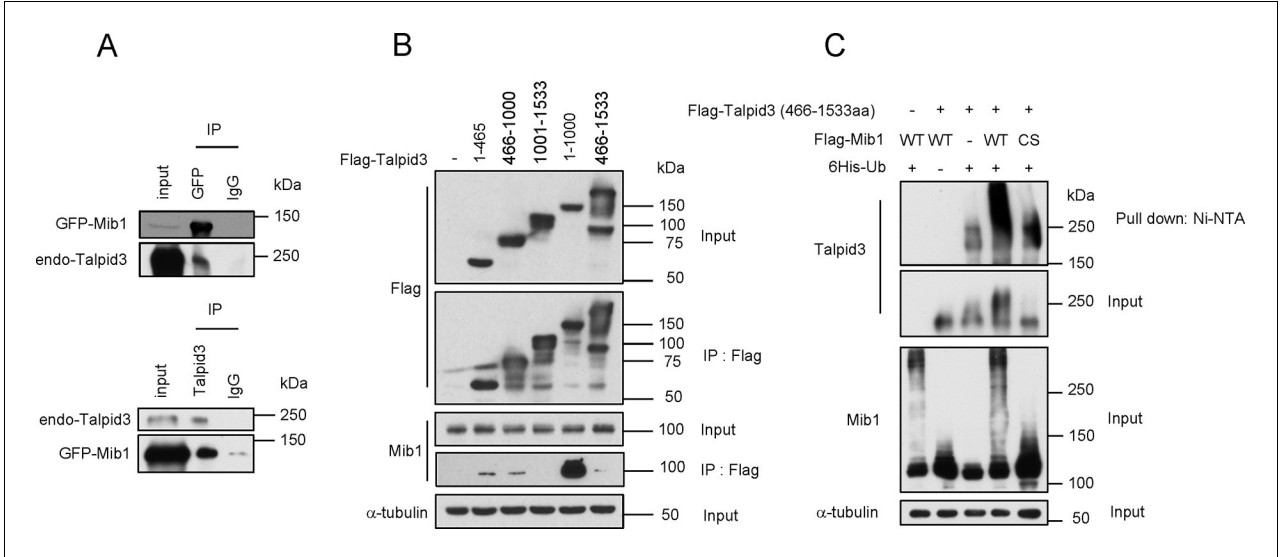

**Figure 6.** Mib1 promotes poly-ubiquitylation of Talpid 3. (**A**) Mib1 interacts with Talpid3. GFP-Mib1 was expressed in HEK293T cells, lysates were subjected to immunoprecipitation with control IgG, anti-GFP, or Talpid3 antibodies, and western blot analysis was performed with the indicated antibodies. (**B**) Mapping of Talpid3 domains that interact with Mib1. HEK293T cells were transfected with control (-) or expression vectors encoding indicated Flag-tagged Talpid3 fragments for 48 h, and lysates were immunoprecipitated with anti-Flag antibodies. The input and immunoprecipitates were analyzed by western blotting with the indicated antibodies. (**C**) Ubiquitylation of Talpid3 depends on the integrity of the Mib1 Ring domain. 293T cells were transfected with plasmids expressing His-ubiquitin, Flag-Mib1-WT, Flag-Mib1-C985S and/or Flag-Talpid3 (466-1533aa) for 2 days, and then purified using Ni-NTA resin to detect in vivo ubiquitylation. The inputs and the Ni-NTA Resin pull-down samples were analyzed by western blotting with the indicated antibodies.

The following figure supplement is available for figure 6:

**Figure supplement 1.** Mib1 ubiquitylates Cep131 and PCM1.

whether ectopic Mib1 expression could promote ubiquitylation of Talpid3 in vivo. We co-expressed Talpid3, with either wild-type or catalytically inactive Mib1 and His-tagged ubiquitin, then affinity purified all ubiquitylated proteins through metal chelation chromatography. We found that Mib1 was highly ubiquitylated, most likely through auto-ubiquitylation, since the catalytically inactive mutant did not exhibit a similar high molecular weight ladder, in line with previous observations (*Figure 6C*; *Villumsen et al., 2013*). More importantly, expression of active Mib1 led to poly-ubiquitylation of Talpid3, suggesting that the ubiquitin ligase activity of Mib1 is required to target this substrate for degradation (*Figures 5D* and *Figure 5—figure supplement 3*).

We tested whether ectopic expression of Mib1 exclusively results in the poly-ubiquitylation of the centriolar protein, Talpid3, or whether it also impacts the ubiquitylation of interacting centriolar satellite proteins, such as Cep131 and PCM1 (*Villumsen et al., 2013*). In contrast with previous studies showing that Mib1 expression resulted in mono-ubiquitylation of Cep131 and PCM1, we found that both proteins were highly poly-ubiquitylated in the presence of active, but not inactive, Mib1 (*Figure 6—figure supplement 1*). These findings suggest that Mib1-mediated poly-ubiquitylation of these targets can promote their destabilization.

## PCM1 sequestration of Mib1 regulates Talpid3 abundance and ciliogenesis

Thus far, we have shown that PCM1 is able to tether Mib1 to centriolar satellites, and the loss of either PCM1 or elevated expression of Mib1 can destabilize targets at satellites (Cep131, PCM1) and centrioles (Talpid3). We reasoned that the aberrant localization of Mib1 brought about by removal of PCM1, its centriolar satellite tether, could perturb the abundance of Talpid3, if indeed the latter protein were a substrate of this E3 ligase. Accordingly, we found that ablation of the *PCM1* gene led to a strong reduction in Talpid3 levels at centrioles (*Figure 7A,B*). Next, we tested

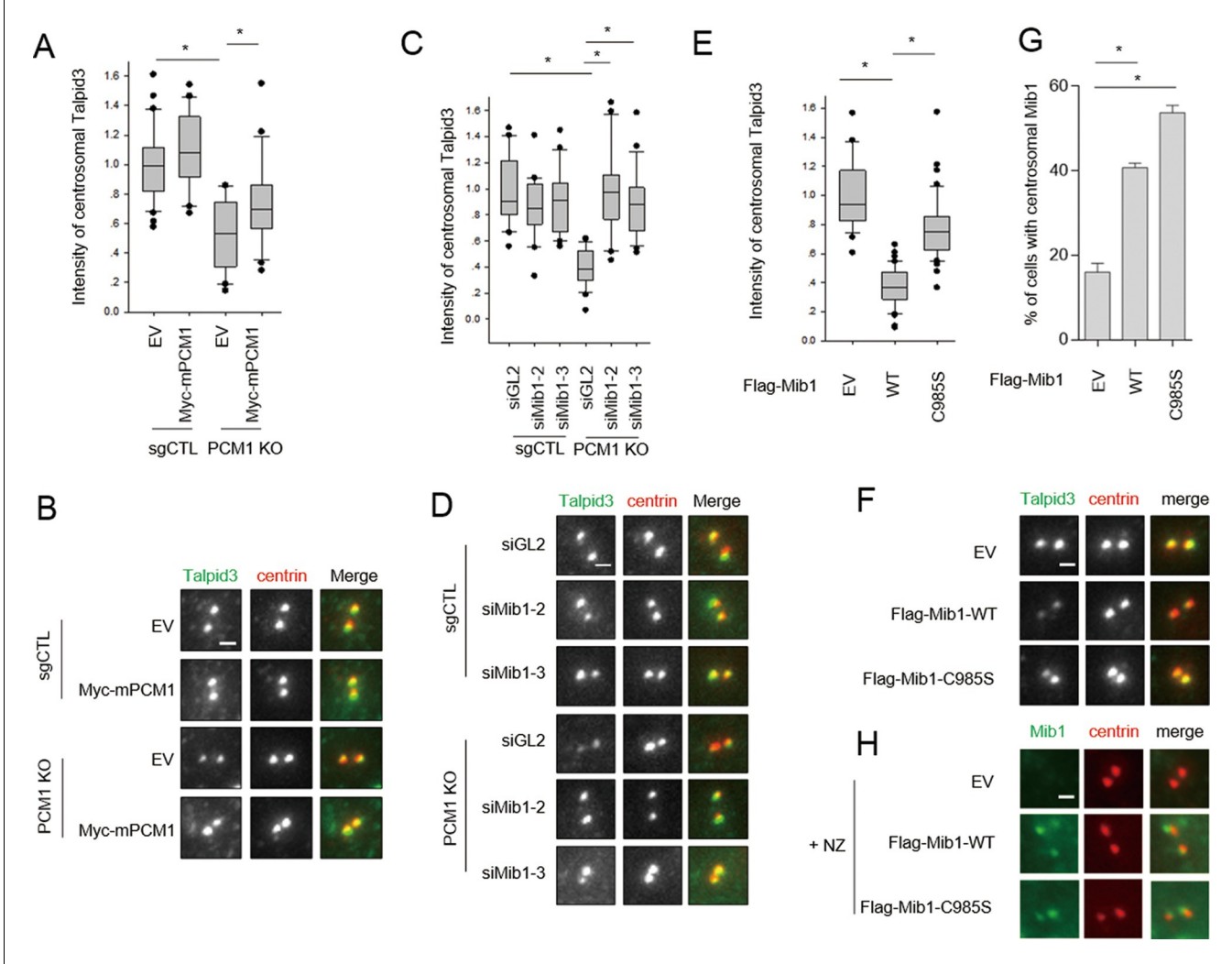

**Figure 7.** PCM1 regulates Talpid3 abundance by sequestering Mib1. (**A**) and (**B**) Ectopic PCM1 expression in *PCM1* KO cells stabilizes Talpid3 at centrioles. Control and *PCM1* KO cells were infected with empty or Myc-mPCM1 lentivirus for 3 days. (**A**) Centrosomal Talpid3 intensity was measured in G1 phase cells by immuno-staining with anti-Talpid3 (green) and centrin (red) antibodies. $N \geq 20$ per sample in two independent experiments. Error bars, SD. *$p<0.05$. (**B**) Representative images are shown. Scale bar, 1 μm. (**C**) and (**D**) Depletion of Mib1 in PCM1 null cells stabilizes Talpid3 at centrioles. Control and *PCM1* KO RPE1 cells were transfected with siGL2, *siMIB1-2* or *siMIB1-3* for 2 days. (**C**) Centrosomal Talpid3 intensity in G1 phase cells was measured by immuno-staining with anti-Talpid3 (green) and centrin (red) antibodies. $N \geq 20$ per sample in two independent experiments. Error bars, SD. *$p<0.05$. (**D**) The representative images are shown. Scale bar, 1 μm. (**E**) and (**F**) Destabilization of Talpid3 at the centrosome by Mib1 is dependent on a functional RING domain. RPE1 cells were infected with empty, Flag-Mib1-WT or Flag-Mib1-C985S lentiviruses for 72 hr. (**E**) Centrosomal Talpid3 intensity in G1 phase cells was measured by immunostaining with Talpid3 (green) and centrin (red) antibodies. $N \geq 20$ per sample in two independent experiments. Error bars, SD. *$p<0.05$. (**F**) The representative images are shown. Scale bar, 1 μm. (**G**) and (**H**) Centrosomal localization of Mib1 in cells ectopically expressing Mib1. (**G**) Cells were analyzed by immuno-staining with Mib1 (green) and centrin (red) antibodies. Cells with centrosomal Mib1 were counted. $N \geq 100$ per sample in two independent experiments. Error bars, SD. *$p<0.05$. (**H**) Representative images are shown. Scale bar, 1 μm.

whether Mib1 sequestration by PCM1 is indeed a key limiting factor in the regulation of Talpid3 levels by depleting Mib1 in *PCM1* knock-out or control cells. Remarkably, silencing Mib1 with either of two siRNAs fully restored wild-type Talpid3 levels (*Figure 7C,D*). Conversely, expression of active, but not inactive, Mib1 likewise drastically reduced the levels of centriolar Talpid3 (*Figure 7E,F*). These studies established for the first time that Mib1, once released from PCM1, can relocate to centrioles and result in the ubiquitylation of Talpid3, promoting its down-regulation.

These experiments, however, did not address whether Mib1 was acting on cytoplasmic pools of Talpid3 or whether it could act locally within the centrosome if it were constitutively transported to that compartment in the presence of wild-type PCM1. To address this question, we fused Mib1 to the PACT domain, which results in transport of linked proteins to the centrosome (*Gillingham and Munro, 2000*). As expected, in the presence of PCM1, the PACT-Mib1 protein localized to centrosomes, whereupon this fusion protein, but not the catalytically inactive mutant, provoked a notable reduction in Talpid3 protein levels (*Figure 8A,B*). In addition, expression of PACT-Mib1, but not the inactive mutant, led to a dramatic reduction in ciliation (*Figure 8C*). Thus, within the limits of our assay, we conclude that Mib1 can act locally at centrioles to de-stabilize Talpid3 (and likely other substrates) and abolish ciliogenesis.

In an effort to establish the epistatic relationships between PCM1, Mib1, and Talpid3 in the cilium assembly pathway, we showed that removal of Mib1 was able to partially restore ciliogenesis after loss of *PCM1*, whereas, Mib1 depletion could not reverse the loss of primary cilia in *TALPID3* ablated cells (*Figure 5B*). These results suggest that Talpid3 acts as an effector of cilium assembly downstream of PCM1 and Mib1.

## PCM1 loss or ectopic Mib1 abrogates early ciliogenic events

We sought a mechanistic explanation for why *PCM1* null cells are unable to assemble cilia. Assembly of this organelle is initiated when mother centrioles mature to form basal bodies and dock ciliary vesicles with the assistance of their distal appendages (*Kim and Dynlacht, 2013*; *Kobayashi and Dynlacht, 2011*; *Sorokin, 1962*). One of the earliest events, detectable within 90 min post-serum deprivation, is the accumulation of Golgi- and endosome-derived Rab8 vesicles at mother centrioles and emergence of a Rab8-positive axoneme. It is known that Rab8 interacts with the distal appendage protein, Cep164 (*Schmidt et al., 2012*). Moreover, Rab8 also associates with the distal centriolar protein, Talpid3, and depletion of Talpid3 abolishes Rab8 localization to basal bodies (*Kobayashi et al., 2014*). We found that Cep164-positive basal bodies were decorated with Rab8 in serum-starved, control cells, as expected (*Figure 9A,B*). In striking contrast, *PCM1* null cells failed to recruit Rab8 to basal bodies. We showed that this defect stemmed from loss of PCM1, since it was rescued by expression of this protein. Importantly, the failure to recruit Rab8 could not be explained by the reduced overall abundance of Rab8 or its upstream regulator, Rab11 (*Figure 1D*). Furthermore, the defect in Rab8 localization was not due to aberrant recruitment of Cep164 (*Figure 9A*), since Cep164 localization was not affected by PCM1 ablation.

Next, we investigated whether diminished Rab8 localization could be brought about through the aberrant re-localization of Mib1 to centrioles in *PCM1* null cells. Remarkably, we showed that expression of the dominant-negative Mib1 could completely restore the recruitment of Rab8, whereas the wild-type protein could not (*Figure 9C,D*). These studies strongly suggest that the arrest of ciliogenesis accompanying PCM1 ablation is due to a failure to accrete Rab8–associated vesicles at the basal body and is a direct result of the mis-localization of Mib1. Further, given that (1) Cep164 localization is unaffected and (2) centriolar Talpid3 levels are sharply reduced in *PCM1* null cells (*Figure 7A–D*), we propose that the striking ciliary defects in these cells stem, at least in part, from enhanced Mib1 localization to centrioles, triggering depletion of Talpid3 and a concomitant loss of Rab8a recruitment.

## Discussion

### Use of *PCM1* null cells to dissect a required role for specific proteins in ciliogenesis

Centriolar satellites are comprised of numerous pro- and anti-ciliogenic proteins, and many of these proteins also localize to centrioles, independently of centriolar satellites (reviewed in *Tollenaere et al., 2015*). This complexity makes it challenging to investigate whether the defects in ciliogenesis resulting from depletion of these proteins can be attributed to centriolar satellite dysfunction, abnormal centriole function, or both. This observation limits our ability to circumscribe the set of centriolar satellite proteins required for ciliogenesis. To this end, we focused on PCM1, as it is an essential organizer of centriolar satellites. By generating *PCM1* knock-out cells and performing structure-function analyses, we found that the pro-ciliogenic function of PCM1 is likely to depend on

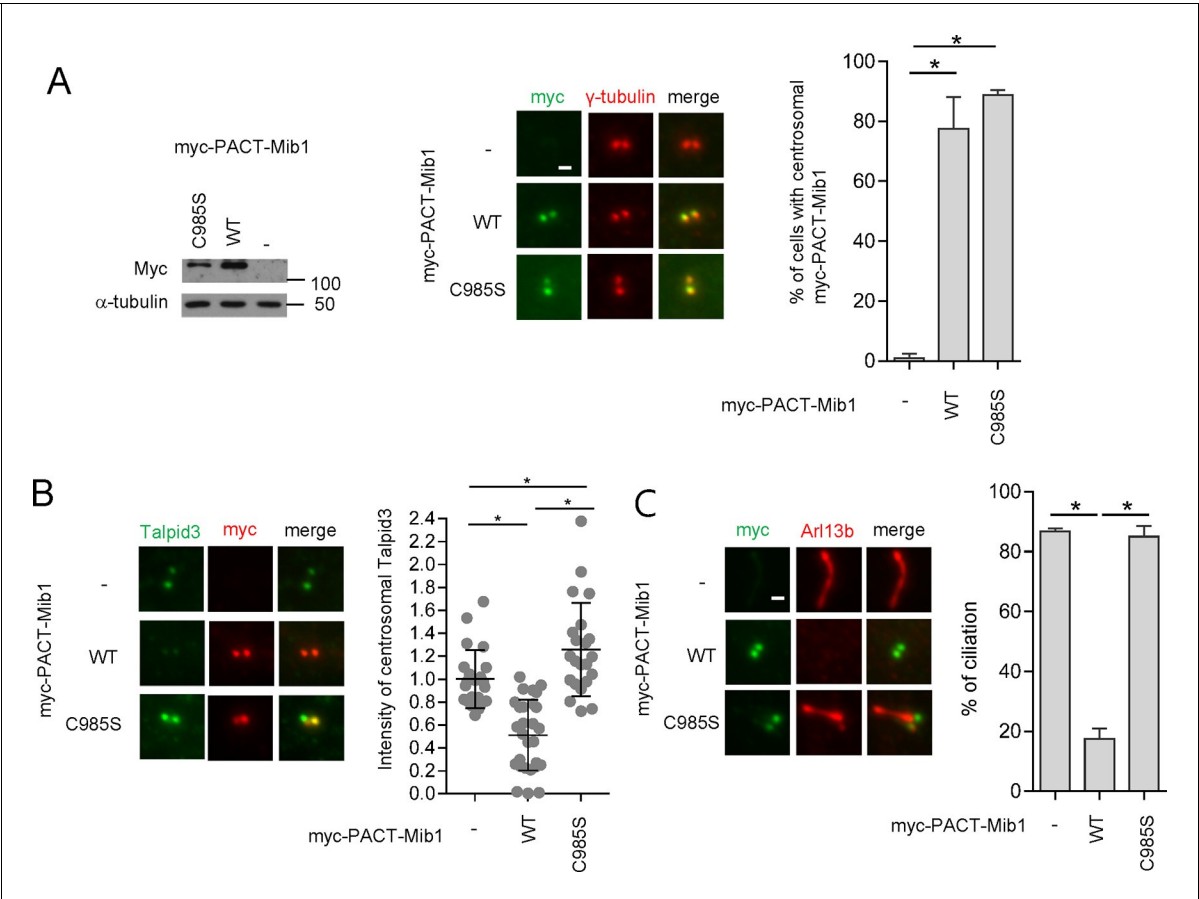

**Figure 8.** Centrosomal targeting of Mib1 destabilizes Talpid3 and inhibits ciliogenesis. (**A**) (*Left*) RPE1 cells were infected with empty (-), myc-PACT-Mib1-WT, or myc-PACT-Mib1-C985S lentiviruses for 72 hr and western blotting performed to detect the fusion protein. (*Middle*) Visualization of centrosomal localization of Mib1 in cells infected as in (A). Myc-PACT-Mib1 was detected by immuno-staining with anti-myc (green) and γ-tubulin (red) antibodies. (*Right*) Cells with centrosomal Mib1 were counted. N ≥ 100 per sample in two independent experiments. Error bars, SD. *p<0.05. (**B**) Destabilization of Talpid3 at the centrosome by centrosomal targeting of Mib1. Centrosomal Talpid3 intensity in G1 phase cells was measured by immuno-staining cells exclusively exhibiting centrosomal myc-PACT-Mib1 with anti-Talpid3 (green) and myc (red) antibodies. N ≥ 40 per sample in two independent experiments. Error bars, SD. *p<0.05. (**C**) Ciliated cells exhibiting centrosomal myc-PACT-Mib1 (as in panel B) were analyzed by immuno-staining with myc (green) and Arl13b (red) antibodies; n≥100 per sample were analyzed in two independent experiments. Error bars, SD. *p<0.05.

its interactions with a limited group of satellite proteins, a subset of which have been analyzed here. Thus, we showed that PCM1-dependent centriolar satellite tethering of Mib1, but not Cep90, is required for ciliogenesis. Use of these tools will accelerate our understanding of the contributions of individual satellite proteins to ciliogenesis through their function at one location or the other. Ongoing proteomic searches for satellite proteins that preferentially interact with the pro-ciliogenic N-terminal domain of PCM1 versus other PCM1 fragments should be most informative.

Cells with defects in cilium assembly frequently exhibit abnormal distributions or morphology of centriolar satellites (*Kim et al., 2008*; *Klinger et al., 2014*; *Kobayashi et al., 2014*; *Sedjaï et al., 2010*; *Stowe et al., 2012*). However, it is not clear whether the distribution and morphology of satellites have a direct impact on ciliogenesis. Specifically, it has been shown that over-expression of BBS4 and Cep72 disrupts centriolar satellite distribution and results in the formation of aggregates that sequester PCM1, which seems to negatively impact ciliogenesis (*Kim et al., 2004*; *Stowe et al., 2012*). We observed similar PCM1-associated foci when we re-introduced PCM1 fragments spanning 1–1200 or 1–1500 into null cells. We found that these foci co-stained with Mib1 and Cep131. Furthermore, these cells ciliated with a frequency comparable to wild-type cells. Our data therefore suggest that the formation of PCM1 foci, particularly those that co-recruit Mib1 and Cep131, does not, per se, inhibit ciliogenesis.

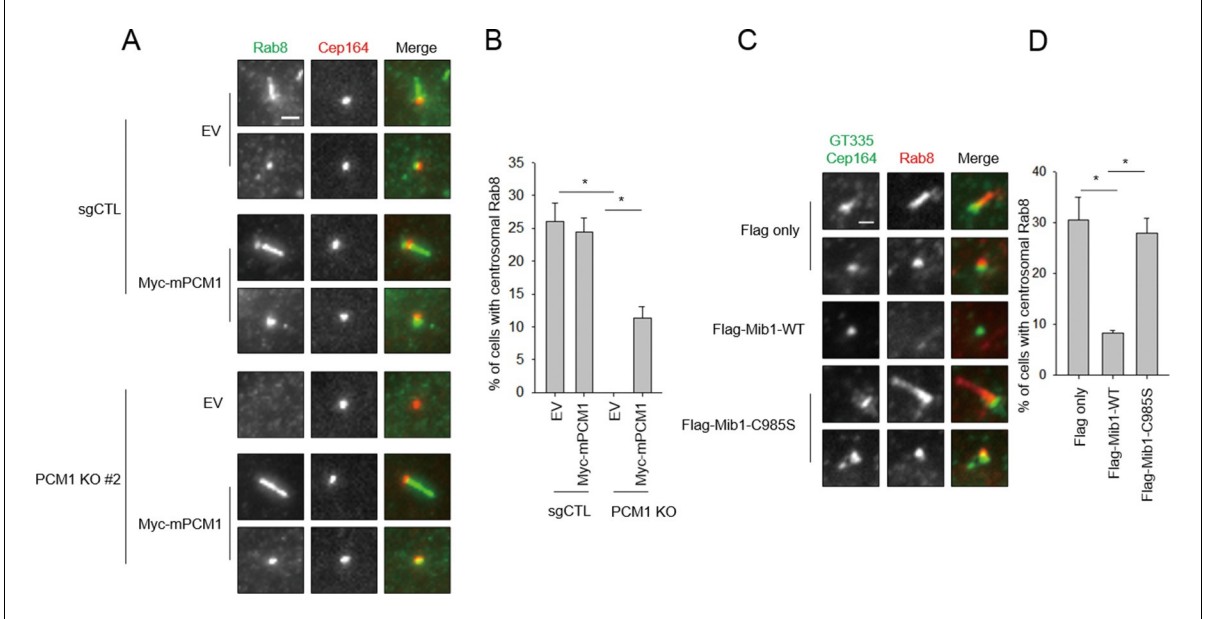

**Figure 9.** Mib1 expression or ablation of PCM1 blocks ciliogenesis at an early stage prior to Rab8 recruitment. (A) PCM1 null cells are defective in recruitment of Rab8 to the mother centriole. (B) Quantitation of Rab8 in *PCM1* null cells. (A) and (B) Control and PCM1 KO RPE1 cells were infected with vector control (EV) or Myc-mPCM1 lentiviruses for 72 hr and serum starved for 6 h, and cells were immuno-stained with Rab8 (green) and Cep164 (red) Representative images are shown. Scale bar, 1 μm. (B) Cells with centrosomal Rab8 signal were counted (n≥100 per sample in two independent experiments). Error bars, SD. *p<0.05. (C) Mib1 E3 ligase activity antagonizes Rab8 recruitment to mother centriole. (D) Quantification of Rab8 recruitment to mother centriole in cells induced to ciliate in the presence of exogenous wild-type or catalytically inactive Mib1. (C) and (D) RPE1 cells were infected with control (Flag only), Flag-Mib1-WT or Flag-Mib1-C985S lentiviruses for 72 hr and serum starved for 6 hr. Cells were immuno-stained with antibodies against Rab8 (red) and Cep164+GT335 (green). N ≥ 100 per sample in two independent experiments. Error bars, SD. *p<0.05.

## A novel model linking PCM1 and its satellite function to an E3 ligase, Talpid3 destruction, and Rab8 recruitment

In this study, we have identified novel roles for the centriolar satellite protein, PCM1, in regulating the localization of an E3 ligase, Mib1, which controls the abundance of key centriolar satellite and centriolar targets. We have elucidated a biochemical pathway linking PCM1 to Mib1, which is able to regulate the abundance of Talpid3—and, most likely several other proteins at centrioles and centriolar satellites–through ubiquitylation, and Rab8 recruitment, which is required for the early stage of ciliogenesis. Although such a linear pathway is the most straightforward model for our observations, other possible explanations cannot be ruled out at this time. For example, it is possible that Mib1 and Talpid3 function through independent pathways and that Talpid3 is required for cilium assembly irrespective of PCM1 status and Mib1 activity. It will therefore be important to expand our view of the Mib1 regulatory network and to identify additional Mib1 targets. Nevertheless, our data vastly extend a model wherein centriolar satellites sequester proteins, such as BBS4, temporally modulating the assembly of the BBSome and the onset of cilium assembly (*Stowe et al., 2012*). BBSome components, including BBS4, have been implicated in cilium assembly, although other studies have not observed such a role for BBS4 (*Yen et al., 2006*; *Mykytyn et al., 2004*; *Hernandez et al., 2013*). Using these *PCM1* null cells, it should now be possible to investigate additional roles for PCM1 in sequestering key components at satellites. It is notable that inactivation of a single protein, Mib1, is able to restore Rab8 recruitment in *PCM1* null cells, as this finding suggests that fine-tuning of Mib1 abundance—and therefore localization–may play a key, rate-limiting step in controlling ciliogenesis.

It is likely that Mib1 will target for destruction other proteins critical for ciliogenesis, in addition to Talpid3. For example, knock-out mice deleted for both isoforms of Rab8 (*Rab8a/Rab8b*) are viable, and it is possible that another Rab (*Rab10*) compensates for the loss of this gene (*Sato et al., 2014*), although the ability of *Rab10* to functionally substitute for Rab8 remains to be shown. It is also

possible that Mib1 could target other components involved in ciliary vesicle assembly. It is also not known whether the related *Mib2* gene can play a role in ciliogenesis. These studies will be accelerated by the use of *Mib1* and *Mib2* knock-out mice, which have been created. These animals display strong Notch phenotypes, similar to zebrafish *Mib1* mutants (*Itoh et al., 2003*; *Kang et al., 2013*; *Koo et al., 2007*). Thus far, ciliary phenotypes have not been observed or examined in these mice, to our knowledge. It would be interesting to examine cilium assembly in these knock-out mice and cells derived from them.

Our work confirms novel findings implicating Mib1 in the regulation of ciliogenesis (*Villumsen et al., 2013*). In this study, Mib1 was shown to interact with PCM1 and Cep131, and, interestingly, induction of cellular stress resulting from UV or heat shock led to the p38 MAPK-dependent inactivation of Mib1, reduced ubiquitylation of Cep131, PCM1, and Cep290, and enhanced ciliogenesis. While our respective studies are in agreement that Mib1 regulates ciliogenesis, our results differ in several significant ways. First, we have observed that each of the centriolar satellite Mib1 substrates (Cep131 and PCM1) are highly poly-ubiquitylated by ectopic Mib1, rather than being mono-ubiquitylated, and this leads to the de-stabilization of both proteins, in contrast with previous work (*Villumsen et al., 2013*). In addition to these centriolar satellite proteins, we show that Talpid3, a centriolar protein required for ciliogenesis, is also poly-ubiquitylated under these conditions. Further, we found that removal of Mib1 led to the stabilization of both Cep131 and PCM1, enhancing the stability of centriolar satellites. We have replicated these findings in two cell lines. At present, we cannot explain the apparent discrepancy between our findings and the previous study and ascribe them to experimental differences (*Villumsen et al., 2013*). We note that a recent study indicated that Mib1 could regulate the stability of another centrosomal protein, Plk4, through lysine 11-, 29-, and 48 ubiquitin linkages (*Čajánek et al., 2015*). Therefore, it should be most interesting to identify additional components regulated by Mib1 and, potentially, to link these findings to the original role of Mib1 in Notch signaling.

It is possible that the loss of satellite integrity could play important roles in human pathological conditions. For example, we note that ~15% of prostate tumors exhibit deletions in the *PCM1* gene (cBioPortal). Given that the loss of cilia is an early event in the transformation of this tissue (*Hassounah et al., 2013*), our studies suggest that further investigation of a role for *PCM1* in prostate tumorigenesis is warranted. One prediction of our model is that Mib1 may be aberrantly up-regulated or mis-localized in these tumors as a result of *PCM1* mutations, which, we have shown, could abolish cilium assembly. In this setting, and perhaps others, loss of *PCM1* and satellite tethering could play a signature role in the transformative process.

# Materials and methods

## Cell culture
Human telomerase-immortalized retinal pigment epithelial cells (hTERT RPE-1, ATCC CRL-4000, passage 3–6) were obtained from ATCC and were confirmed to be free of mycoplasma. Cells were grown in DMEM containing 10% FBS and antibiotics (penicillin and streptomycin). To induce cilia formation, RPE1 cells were placed in DMEM without FBS for 24 or 48 hr.

## Transfection
siRNAs were transfected into RPE1 cells using RNAiMAX (Invitrogen, Carlsbad, CA ) according to the manufacturers' manual. Plasmids were transfected into 293T cells by polyethylenimine.

## siRNA sequences
siRNAs used in this research were *siMIB1-1* (5'-GACUGAUGGAAUGUUUGAGACUUUA-3') (*Berndt et al., 2011*), *siMIB1-2* (5'-GCATATGTCCTCTGGGATAUU-3'), *siCTL* (5'-UUCUCCGAACG UGUCACGUUU-3'), *siGL2*(5'-CGUACGCGGAAUACUUCGAUU-3') and *siPCM1* (5'-GGCUUUAAC UAAUUAUGGAdTdT-3') (*Villumsen et al., 2013*).

## DNA constructs
Mouse/human PCM1 (*Kim et al., 2012*) and truncation mutants were sub-cloned into pLVX-IRES-Puro (Clonetech, Mountain View, CA) with a Myc-tag, Human Cep131 and human Mib1 were kindly

provided by Kunsoo Rhee (Seoul National University, Seoul, South Korea) and Randall T. Moon (University of Washington School of Medicine, Seattle, USA), respectively. Mib1 and PACT-Mib1 were sub-cloned into PLVX-IRES-Puro with Flag or myc tag. Mib1 (C985S) was generated by site-direct mutagenesis using PCR. Talpid3 (*Itoh et al., 2003*), and Mib1 deletions were sub-cloned into (pCMV5-Flag).

## Antibodies

Antibodies used included: Mib1 (M5948; Sigma-Aldrich, St. Louis, MO), PCM1 (H262; Santa Cruz, Santa Cruz, CA), Cep131 (A301-415A; Bethyl Laboratories, Montgomery, TX), Cep90 (*Kim and Rhee, 2011*), Ki-67 (ab15580; Abcam, Cambridge, MA), GT335 (AG-20B-0020-C100; Adipogen, Switzerland), BBS4 (*Kim et al., 2012*), Rab11 (71–5300; Life Technologies, Carlsbad, CA), Rab8 (a gift from J. Peranen, University of Helsinki, Helsinki, Finland and 610844; BD Biosciences, San Jose, CA), α-tubulin (T5168; Sigma-Aldrich), Cep290 (A301-659A; Bethyl Laboratories), Ofd1 (a gift from J. Reiter, University of California, San Francisco, USA; *Singla et al., 2010*), Talpid3 (*Kobayashi et al., 2014*), Myc (sc-40; Santa Cruz), Flag (F3165; Sigma-Aldrich), centrin (04–1624; Millipore, Billerica, MA), GFP (G1544; Santa Cruz), Cep164 (a gift from Eva Lee, University of California, USA), IFT88 (13967-1-AP; Proteintech, Chicago, IL).

## Immunofluorescence, microscopy, and statistical analyses

RPE1 cells were grown on a coverslip and fixed with cold methanol for 10 min or 4% PFA form 15 min. The cells were incubated with 3% BSA-0.3% PBST (Triton X-100) for 20 min. Primary and secondary antibodies in the blocking solution were placed on the cells for 2 hrs at RT Antibodies were diluted in 0.1% PBS with Triton X-100 with 3% BSA. The cells were mounted onto a slide glass with mounting solution (P36934; Life Technologies) and imaged using a microscope (63× or 100×, NA 1.4; Axiovert 200M, Carl Zeiss, Germany) equipped with a cooled CCD (Retiga 2000R; QImaging, Surrey, Canada) and MetaMorph . The images were obtained as z projections (0.3 µm interval) and analyzed by ImageJ (NIH, http:// rsbweb.nih.gov/ij/) and SigmaPlot (Systat Software, San Jose, CA). The region of interest was defined by drawing a circle including the centrosome. Background values were measured from the same-sized circle as a circle including the centrosome in an adjacent region. Talpid3 staining was analyzed in G1 phase cells (serum starved for 24 or 48 hr) to achieve uniformity and to avoid oscillations in abundance during the cell cycle.

## In vivo ubiquitination assay

293 cells were co-transfected with a plasmid encoding His-Ub (gift from R. Baer), Flag-Mib1 (WT or C985S), and a plasmid encoding substrates (Flag-Talpid3, Myc-mPCM1 or GFP-Cep131) for 48 hrs. The cells were lysed in Buffer A (100 mM NaH$_2$PO$_4$, 10 mM Tris-HCl, 6 M guanidine-HCl, 10 mM imidazole pH 8.0) and sonicated. The cell lysates were incubated with nickel-NTA resin (Qiagen) for 3 hr at RT. The beads were washed with buffer A twice, bufferA/TI (1:1 ratio) (TI buffer, 25 mM Tris-Cl pH 6.8 and 20 mM imidazole), and TI buffer. And then the beads were incubated with SDS loading buffer containing 200 mM imidazole and boiled for 10 min.

## Lentivirus production and infection

For lentivirus production, 293T cells were transfected with lentiviral envelope, packaging and expression vectors for 2 days. And then the media were harvested and filtered. For infection, RPE1 cells were incubated with the media containing lentivirus and polybrene (8 or 12 µg/ml) for 24 hr.

## Immunoprecipitation

293T cells were lysed by ELB buffer (50 mM Hepes pH7, 150 mM NaCl, 5mM EDTA pH 8, 0.1% NP-40, 1 mM DTT, 0.5 mM AEBSF, 2 µg/ml leupeptin, 2 µg/ml aprotinin, 10 mM NaF, 50 mM β-glycerophosphate, and 10% glycerol) on ice for 10 min and the lysates were centrifuged at 14,000 rpm and 4°C. The supernatant was incubated with Myc antibody - Protein G Sepharose (GE Healthcare) or Flag beads (Sigma-Aldrich). For immunoprecipitation, 2 mg of the resulting supernatant after centrifugation was incubated 4°C. The beads was washed with ELB buffer four times, boiled with SDS loading buffer, and analyzed by immuno-blotting.

## Establishment of knock-out cell lines using CRISPR/Cas9

RPE1 cells were infected with lentivirus expressing Flag-Cas9 and sgRNA and grown for 10 days, after which the cells were separated as single cells into 96 well plates. After 2 weeks, the colonies were analyzed for genomic editing. sgRNAs used included: sgPCM1 (5'-ATGATCAGGATTTAC-CAAAC-3') and sgCTL (5'-GAGACGTCTAGCACGTCTCT-3'). For Talpid3 targeting, the following sgRNA was used: 5'-GATGATGTTCTTCATGACCT-3'.

## Statistics and reproducibility

The statistical significance of the difference between two means was determined using a two-tailed Student's t-test. All data are presented as mean ± SEM or SD as specified in the figure legends. Differences were considered significant when $p < 0.05$. Results reported are from three or two independent biological replicates as noted in legends with reproducible findings each time. Exact means, SEM, SD, N, P values can be found in the *Supplementary file 1*.

## Acknowledgements

We thank the members of the Dynlacht laboratory for many fruitful discussions. We thank S Lau for help in quantitation of the immunofluorescence microscopy. We thank J Peranen, R Moon, K Rhee, J Reiter, S Shi and E Lee for critical reagents. This work was supported by NIH grant 1R01HD069647 to BDD.

## Additional information

### Funding

| Funder | Grant reference number | Author |
|---|---|---|
| National Institute of Child Health and Human Development | 1R01HD069647 | Brian D Dynlacht |

The funders had no role in study design, data collection and interpretation, or the decision to submit the work for publication.

### Author contributions

LW, KL, Conception and design, Acquisition of data, Analysis and interpretation of data, Drafting or revising the article; RM, Acquisition of data, Analysis and interpretation of data; IS, Conception and design, Drafting or revising the article; BDD, Conception and design, Analysis and interpretation of data, Drafting or revising the article

### Author ORCIDs

Brian D Dynlacht, http://orcid.org/0000-0001-9485-512X

## Additional files

### Supplementary files

• Supplementary file 1. Statistical information.

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
