## [Decision Letter]

[Editors’ note: this article was originally rejected after discussions between the reviewers, but the authors were invited to resubmit after an appeal against the decision.]

Thank you for submitting your work entitled "Tethering of an E3 ligase by PCM1 regulates the abundance of centrosomal Talpid3 and promotes ciliogenesis" for consideration by *eLife*. Your article has been reviewed by two peer reviewers, and the evaluation has been overseen by a Reviewing Editor and Vivek Malhotra as the Senior Editor. Our decision has been reached after consultation between the reviewers. Based on these discussions and the individual reviews below, we regret to inform you that your work will not be considered further for publication in *eLife*.

The reviewers found the manuscript contained a significant amount of high quality data (like the PCM1 KO phenotypes) that could constitute an important contribution to the field. However, the proposed model does not seem to fit together as outlined below and the in vivo function of Mib1 seems at odds with this work as pointed out by the second reviewer. Based on the comments of the experts, we cannot proceed further with your manuscript for consideration at *eLife*. Should you manage to obtain experimental support for the issues raised by the reviewers, we would be willing to consider a new manuscript for a full review. In any case, we hope you will find our reviewers comments helpful in strengthening your proposal for submission elsewhere or as mentioned above to *eLife* as a new manuscript.

The major issues follow.

The model pathway of PCM1 controlling MIB1, which in turns controls TALPID3 is not fully supported by the data and that other models are not considered.

Issues around the model include: a) The model suggests that MIB1 is a critical regulator of cilia formation, which is at odds with the mouse knock-out phenotype as pointed out by reviewer 2.b) TALPID3 levels do not appear to be affected in RPE1 cells with altered MIB1 (Figure 5). The data in U2OS cells is better, but there are no cilia formation assays with these cells. c) The alternative interpretation of the epistasis relationship between MIB1 and TALPID3 is that they are on independent pathways and that TALPID3 is required for cilia formation regardless of PCM1/MIB1 status. d) A further complication is that your group reported that TALPID3 depletion causes defects in centriolar satellites (Kobayashi et al., 2014, JCB 204:215-229). What is the status of MIB1 in the TALPID3 KO cells?

Additionally, the discussion of Villumsen et al. (2013, EMBO J. 32:3029-3040) seems dismissive. This study should be discussed in more detail.

Reviewer #1:

This paper examined the function of PCM1 by knocking it out in RPE1 cells. The paper has a lot of nice, convincing data but the paper is poorly written and over interpretation is rampant. No alternatives are considered for experiments and the possibility that additional proteins may participate in the phenotypes is not considered.

Most of my concerns could be addressed with a careful rewrite of the manuscript. However, a major conclusion of the paper is that the loss of PCM1 allows Mib1 to relocate to the centrioles and destabilize a number of proteins thus preventing ciliogenesis. The support for this happening at the centriole is weak. For this conclusion to be made, the authors need to relocalize Mib1 to centrioles even when PCM1 is present and show that this has the same phenotype as loss of PCM1.

"These results lead to several important conclusions. First, PCM1 promotes ciliogenesis through interactions with a specific cohort of proteins, including Mib1, OFD1, Cep131, Cep72 and Cep290…". No evidence that is true. It could be one of these proteins or it could be something else.

Subsection “PCM1 anchors Mib1 at centriolar satellites”: "because the PCM1-Mib1 interaction is essential for restoration of ciliogenesis in PCM1 knock-out cells (Figure 2)." This was not demonstrated.

In the same section: "Moreover, ectopic expression of Mib1 promoted its association with the centriolar protein Talpid3, indicating re-localization to centrioles (Figure 6) and suggesting that the number of binding sites for Mib1 (within centriolar satellites) may be saturable." Nothing in Figure 6 shows relocalization to centrioles.

Subsection “PCM1 regulates ciliogenesis through sequestration of Mib1”: "Expression of Mib1 led to the near-complete dissolution of centriolar satellites typified by PCM1 enrichment, whereas inactive Mib1 dramatically led to their stabilization (Figure 5—figure supplement 3)." Figure 5—figure supplement 3 shows that PCM1 is mostly gone when Flag-Mib1-WT is expressed. Thus one cannot say anything about satellites using PCM1 as the marker.

In the same section: "Consistent with this observation, depletion of Mib1 with two distinct siRNAs led to dramatic up-regulation of centriolar satellites, concomitant with Cep131 and PCM1 stabilization (Figure 5—figure supplement 2).” There is nothing in Figure 5—figure supplement 2 to address this point. Figure 5—figure supplement 3 shows western blots to support the increase in Cep131 and PCM1, but I cannot find the data to support the up-regulation of centriolar satellites.

Subsection “PCM1 sequestration of Mib1 regulates Talpid3 abundance and ciliogenesis”: "These studies established for the first time that Mib1, once released from PCM1, can relocate to centrioles and result in the ubiquitylation of Talpid3, promoting its down regulation." No evidence that Mib1 is acting at the centrioles; it could be acting in the cytoplasm.

Figure 2: the BBS4 mapping is not convincing. This needs to be improved or removed.

Legend to Figure 2: "We surmise that the Cep131 band in cells expressing fragment 1-800 is cross-reactive and reproducibly migrates with higher mobility than native Cep131." I have no idea what this means.

The reported aberrant localization of Mib1 and Cep131 to centrioles in PCM1 KO cells is not convincing (Figure 3). It looks like these proteins are at the centrioles in the controls. These need to be examined in nocodazole treated cells as was done in Figure 4 agree that the 1-1200 fragment is displacing them from the centriole region, the foci no longer look like satellites so it is not clear what this means.

Figure 3, the failure to localize Cep90 to satellites could just be a consequence of its level in the cell. Figure 1 shows that it is strongly reduced in the absence of PCM1.

Reviewer #2:

The function of centriolar satellites, which form a dispersed halo around the centriole, has been the topic of extensive speculation but has not been analyzed rigorously. This manuscript presents a solid set of data that demonstrate the importance of the pericentriolar satellite protein PCM1 in formation of cilia in RPE1 cells. Previous knockdown experiments had shown a mild effect of PCM1 deletion on ciliogenesis. Here, the investigators make null mutant cells using CRISPR/Cas and document the complete absence of cilia in mutant cells. They also show that PCM1 is required for the focal localization in satellites of all tested PCM proteins, demonstrating the central role of PCM1 in these structures.

The investigators show that the phenotypes can be rescued by expression of full-length PCM1 and then use standard structure-function experiments to identify regions of PCM1 that are required to rescue the ciliogenesis defect and that bind to other satellite proteins.

The authors then focus on Mib1, an E3 ubiquitin ligase, which was recently shown by another group to be enriched in centriolar satellites and suggested that Mib1 can suppress cilia formation. The authors show Mib1 and some other satellite proteins relocalize from satellites to centrioles in the PCM1 KO cells. They go on to show that siRNA knockdown of Mib1 in the PCM1 KO cells restores some ciliogenesis (to about half the level in control cells); this is an important and interesting result that shows that PCM1 promotes ciliogenesis, at least in part, by removing Mib1 from the centriole. It would be helpful to show some images of these cilia to demonstrate whether these cilia are normal in structure and length. In a supporting experiment (Figure 5), the authors show that overexpression of wild-type Mib1, but not a catalytically inactive form of Mib1, inhibits cilia formation; the text describing this experiment is confusing and should be rewritten.

The authors then investigate what substrate of Mib1 at the centriole could regulate cilia formation and show that several centriolar and satellite proteins can be polyubiquitylated by Mib1. They focus on Talpid3, which is puzzling, as they state that there is no direct interaction between Mib1 and Talpid3 and that the proteins normally reside in different compartments, and the in vivo interactions are therefore likely to be indirect, and the effects on Talpid3 levels could be mediated indirectly by other Mib1 targets.

In a final set of experiments, the authors show that Rab8 fails to localize to the centriole in PCM1 mutant cells and that some Rab8 can accumulate at the centriole when Mib1 is overexpressed, consistent with the hypothesis that sequestration of Mib1 away from the centriole by PCM1 allows Rab8 to be recruited to the mother centriole in an early stage of ciliogenesis. This logic of a PCM1-Mib1-Talpid3-Rab8 pathway is reasonable, but its in vivo relevance not entirely compelling, as there is no in vivo evidence for the role of Mib1 in cilia formation (Mib1 null mice show phenotypes consistent with loss of Notch signaling) and Rab8a null mutants are viable, which should be acknowledged by the authors.

Overall, the analysis of the PCM1 null phenotype and its relationship to Mib1 is intriguing and important.

[Editors’ note: what now follows is the decision letter after the authors submitted for further consideration.]

Thank you for resubmitting your work entitled "Tethering of an E3 ligase by PCM1 regulates the abundance of centrosomal KIAA0586/Talpid3 and promotes ciliogenesis" for further consideration at *eLife*. Your revised article has been favorably evaluated by Vivek Malhotra (Senior editor), a Reviewing editor, and one reviewer.

The Reviewing editor agrees with the reviewer that the authors have diligently and completely addressed the comments found in the previous reviews. Particularly important are the experiments with Mib1 fused to the PACT domain to force localization of Mib1 to centrosomes. This construct behaves as predicted and leads to reduced levels of Talpid3. The manuscript has been substantially improved but there are some remaining minor issues that need to be addressed before acceptance, as outlined below:

A few minor revisions are necessary to address the published work cited by the reviewer. The reviewer lists three papers to cite and whose citation will require some modification to the text.

Reviewer #1:

The authors have done an excellent job of responding to my criticisms and feel that the new experiments and rewording have addressed my concerns. However, the new Discussion has some issues that need to be addressed.

Double Mib1-/-, Mib2-/- mouse knockouts have been made. Mib2-/- does not have much of a phenotype while the Mib1-/-, Mib2-/- look like the Mib1-/- animals (PMID: 18043734). Since the loss of Mib2 does not alter the Mib1 phenotype, the compensation argument in the mouse is weak. In addition, for the authors to make this argument, they need to show that Mib2 is not normally expressed in their cells and that the ectopic expression of Mib2 rescues the loss of Mib1.

The Rab8 discussion is superficial as there are two Rab8 isoforms. The knockout of both has little effect in the mouse (PMID: 24213529). Rab10 may be compensating but I don't know of any work that shows that Rab10 comes in to fill the role of Rab8 when Rab8a/b are missing.

The BBS4 discussion is not correct as mice and other organisms lacking this gene still form cilia (PMID: 15173597).

---

## [Author Response]

[Editors’ note: the author responses to the first round of peer review follow.]

The reviewers found the manuscript contained a significant amount of high quality data (like the PCM1 KO phenotypes) that could constitute an important contribution to the field. However, the proposed model does not seem to fit together as outlined below and the in vivo function of Mib1 seems at odds with this work as pointed out by the second reviewer. Based on the comments of the experts, we cannot proceed further with your manuscript for consideration at eLife. Should you manage to obtain experimental support for the issues raised by the reviewers, we would be willing to consider a new manuscript for a full review. In any case, we hope you will find our reviewers comments helpful in strengthening your proposal for submission elsewhere or as mentioned above to eLife as a new manuscript.

The major issues follow.

The model pathway of PCM1 controlling MIB1, which in turns controls TALPID3 is not fully supported by the data and that other models are not considered.

Issues around the model include: a) The model suggests that MIB1 is a critical regulator of cilia formation, which is at odds with the mouse knock-out phenotype as pointed out by reviewer 2.

In our paper, we show that Mib1 is a negative regulator of ciliogenesis based on data in RPE1 cells, consistent with a previous report showing that knock-down of Mib1 promotes ciliogenesis in human RPE1 cells (Villumsen et al. 2013). In published papers reporting the Mib1 null mice, however, ciliogenesis was not studied. Indeed, the dominant phenotype in Mib1 knockout mice reflects defects in Notch signaling. It is not known whether MIB1-/- MEFs have a higher ciliation rate compared to wild-type cells, but since this has not been examined, it is not accurate to state that the knock-out study disagrees with our work. Apart from this fact, it is also well known that mutants often show compensation (and there is a second Mib gene, Mib2). This point is further driven home by the Rab8 knock-out data cited by Reviewer 2. These animals are indeed viable, as we point out in our revised manuscript, but they are viable most likely because there is compensation by other Rab family members. Thus, additional mouse models will be needed in the future to address the role of Mib1 (and, potentially, Mib2) in ciliation and development.

b) TALPID3 levels do not appear to be affected in RPE1 cells with altered MIB1 (Figure 5). The data in U2OS cells is better, but there are no cilia formation assays with these cells.

We believe that key data were overlooked here. First, we showed that ablation of the PCM1 gene led to a strong reduction in Talpid3 levels at centrioles, specifically (Figure 7), and this coincides with the relocation of Mib1 from centriolar satellites to centrioles. We also definitively show that over-expression of Mib1 results in a decrease in Talpid3 levels on centrioles in RPE1 cells, while knock-down of Mib1 could rescue Talpid3 levels on centrioles in PCM1-/- cells (Figure 7). However, the total, steady-state levels of Talpid3 were not affected in PCM1 depleted cells (Figure 5), strongly reinforcing our model that Mib1 regulation of Talpid3 occurs on centrioles. We have now performed additional experimentation in response to this critique by fusing Mib1 to the PACT domain, which targets proteins to centrosomes. These data, in new Figure 8, definitively show that Talpid3 levels are indeed reduced at centrioles, strengthening our prior conclusions regarding Talpid3 regulation at centrosomes by Mib1.

Parenthetically, it is worth pointing out that U2OS cells do not ciliate, so ciliation could not be tested in these cells.

c) The alternative interpretation of the epistasis relationship between MIB1 and TALPID3 is that they are on independent pathways and that TALPID3 is required for cilia formation regardless of PCM1/MIB1 status.

Using standard epistasis-type experiments in mammalian cells, we provide several lines of evidence to demonstrate that PCM1, Mib1, and Talpid3 are in the same pathway during ciliogenesis: (1) Ablation of PCM1 leads to the translocation of Mib1 from centriolar satellites to centrioles and decreased Talpid3 protein levels on centrioles; (2) Mib1 interacts with Talpid3 and promotes poly-ubiquitylation of Talpid3; (3) over-expression of Mib1 results in defective ciliogenesis, whereas knock-down of Mib1 rescues the ciliation defect in PCM1-/- cells; (4) over-expression of Mib1 (with or without an attached PACT domain directing the protein to centrosomes) results in decreased Talpid3 on centrioles, whereas knock-down of Mib1 rescues Talpid3 levels on centrioles in PCM1-/- cells. Therefore, the most straightforward interpretation is to place these proteins in a linear pathway, as we have done. Nevertheless, it is formally possible that an alternative pathway exists, and we have now revised our Discussion to incorporate this idea.

*d) A further complication is that your group reported that TALPID3 depletion causes defects in centriolar satellites (Kobayashi* et al.*, 2014, JCB 204:215-229). What is the status of MIB1 in the TALPID3 KO cells?*

Based on the totality of our data described above, Mib1 is an upstream regulator of Talpid3, and we do not expect Mib1 to be de-regulated in Talpid3 KO cells. However, to address this point, we have now examined Mib1 levels in Talpid3 KO cells. As anticipated from our prior work, PCM1 and centriolar satellites are more highly concentrated in the Talpid3 KO cells. In the Talpid3 KO cells, Mib1 localizes normally on the centriolar satellites without going to centrioles. Please see new Figure 5—figure supplement 2 (Discussion section). We thank the Editor for prompting us to include these new data.

*Additionally, the discussion of Villumsen* et al. *(2013, EMBO J. 32:3029-3040) seems dismissive. This study should be discussed in more detail.*

We make several references to this work. In addition, we have modified and vastly expanded the discussions in several sections to more fully discuss the results of this paper.

Reviewer #1:

*This paper examined the function of PCM1 by knocking it out in RPE1 cells. The paper has a lot of nice, convincing data but the paper is poorly written and over interpretation is rampant. No alternatives are considered for experiments and the possibility that additional proteins may participate in the phenotypes is not considered. Most of my concerns could be addressed with a careful rewrite of the manuscript. However, a major conclusion of the paper is that the loss of PCM1 allows Mib1 to relocate to the centrioles and destabilize a number of proteins thus preventing ciliogenesis. The support for this happening at the centriole is weak. For this conclusion to be made, the authors need to relocalize Mib1 to centrioles even when PCM1 is present and show that this has the same phenotype as loss of PCM1.*

We thank the reviewer for his/her constructive comments. The reviewer is concerned that we have not provided sufficiently strong support for the conclusion that loss of PCM1 causes Mib1 to relocate to centrioles, and he/she suggested that we perform an experiment in which we re-localize Mib1 to centrioles in cells with normal PCM1 levels. We have performed this experiment and now provide these data in new Figure 8. Specifically, we localized Mib1 to centrioles by expressing a PACT-Mib1 fusion protein in RPE1 cells, as this will constitutively target Mib1 to centrosomes, and examined Talpid3 levels on centrioles decorated with the PACT-Mib1 protein. As shown in Figure 8, there was a significant diminution in Talpid3 levels on centrioles in these cells, whereas the catalytically inactive Mib1 C985S mutant showed slightly elevated Talpid3 levels. Furthermore, PACT-Mib1 expression also led to strong inhibition of ciliogenesis. These results are discussed in subsection “PCM1 sequestration of Mib1 regulates Talpid3 abundance and ciliogenesis” of the manuscript. PACT-Mib1 expression did not result in complete loss of Talpid3, most likely owing to limiting quantities of proteasomes within the centrosomal compartment.

We would like to emphasize that, apart from these new data, several other lines of evidence also support our conclusion. First, ablation of PCM1 leads to the translocation of Mib1 from centriolar satellites to centrioles and decreased Talpid3 protein levels on centrioles (Figure 3, Figure 4, and 7 and Figure 2—figure supplement 1). Second, Mib1 interacts with Talpid3 and promotes poly-ubiquitylation of Talpid3 (Figure 6). Third, over-expression of Mib1 results in defective ciliogenesis, whereas knock-down of Mib1 partially rescues the ciliation defect in PCM1-/- cells (Figure 5). Finally, over-expression of Mib1 (with or without an attached PACT domain) results in decreased Talpid3 on centrioles, whereas knock-down of Mib1 rescues Talpid3 levels on centrioles in PCM1-/- cells (Figure 7 and new Figure 8).

Altogether, we believe that our data strongly support the conclusion that loss of PCM1 allows Mib1 to relocate to centrioles where it destabilizes Talpid3 and other proteins required for ciliogenesis.

"These results lead to several important conclusions. First, PCM1 promotes ciliogenesis through interactions with a specific cohort of proteins, including Mib1, OFD1, Cep131, Cep72 and Cep290…". No evidence that is true. It could be one of these proteins or it could be something else.

We agree and have modified this sentence accordingly.

Subsection “PCM1 anchors Mib1 at centriolar satellites”: "because the PCM1-Mib1 interaction is essential for restoration of ciliogenesis in PCM1 knock-out cells (Figure 2)." This was not demonstrated.

We have modified the sentence to: “… the domain in PCM1 essential for ciliogenesis can interact with Mib1”.

In the same section: "Moreover, ectopic expression of Mib1 promoted its association with the centriolar protein Talpid3, indicating re-localization to centrioles (Figure 6) and suggesting that the number of binding sites for Mib1 (within centriolar satellites) may be saturable." Nothing in Figure 6 shows relocalization to centrioles.

We have now ectopically expressed Mib1 in RPE1 cells and examined its relocation to centrioles in the presence of nocodazole to remove satellite staining. Please see new Figure 7.

Subsection “PCM1 regulates ciliogenesis through sequestration of Mib1”: "Expression of Mib1 led to the near-complete dissolution of centriolar satellites typified by PCM1 enrichment, whereas inactive Mib1 dramatically led to their stabilization (Figure 5—figure supplement 3)." Figure 5—figure supplement 3 shows that PCM1 is mostly gone when Flag-Mib1-WT is expressed. Thus one cannot say anything about satellites using PCM1 as the marker.

The reviewer is correct, and we thank him/her for the suggestion. Indeed, we have examined other satellite markers (including Cep131 and Cep290) and confirmed the dissolution of satellites. We now show the data for Cep131 and Cep290, and cells express essentially normal total levels of the latter protein when Flag-Mib1-WT is expressed. Please see new Figure 5—figure supplement 3, which strengthen our conclusion stated previously.

In the same section: "Consistent with this observation, depletion of Mib1 with two distinct siRNAs led to dramatic up-regulation of centriolar satellites, concomitant with Cep131 and PCM1 stabilization (Figure 5—figure supplement 2).” There is nothing in Figure 5—figure supplement 2 to address this point. Figure 5—figure supplement 3 shows western blots to support the increase in Cep131 and PCM1, but I cannot find the data to support the up-regulation of centriolar satellites.

We apologize for the typographical mistake, since the data we cited is in Figure 5—figure supplement 3, not Figure 5—figure supplement 2. Figure 5—figure supplement 3 shows enhanced PCM1 staining on centriolar satellites. As described above, we now show immunofluorescence data for PCM1, Cep290, and Cep131 in new Figure 5—figure supplement 3.

Subsection “PCM1 sequestration of Mib1 regulates Talpid3 abundance and ciliogenesis”: "These studies established for the first time that Mib1, once released from PCM1, can relocate to centrioles and result in the ubiquitylation of Talpid3, promoting its down regulation." No evidence that Mib1 is acting at the centrioles; it could be acting in the cytoplasm.

The reviewer is correct that we did not rule out the possibility that Mib1 could act on cytoplasmic pools (it is in fact challenging to demonstrate this under any circumstance). However, as described above, we have now targeted Mib1 to centrioles when PCM1 is present by ectopically expressing PACT-Mib1 in cells expressing PCM1. Please see new Figure 8. These data confirm that Mib1 can act locally on Talpid3 at centrioles when directed specifically to that compartment.

Figure 2: the BBS4 mapping is not convincing. This needs to be improved or removed.

In response to this comment, we have removed the BBS4 data from this figure.

Legend to Figure 2: "We surmise that the Cep131 band in cells expressing fragment 1-800 is cross-reactive and reproducibly migrates with higher mobility than native Cep131." I have no idea what this means.

We apologize for the confusion and have now clarified the sentence.

The reported aberrant localization of Mib1 and Cep131 to centrioles in PCM1 KO cells is not convincing (Figure 3). It looks like these proteins are at the centrioles in the controls. These need to be examined in nocodazole treated cells as was done in Figure 4 agree that the 1-1200 fragment is displacing them from the centriole region, the foci no longer look like satellites so it is not clear what this means.

We have now examined Cep90, Cep131 and Cep290 in both nocodazole-treated and untreated cells (new Figure 1—figure supplement 1) and found that these proteins localize on both satellites and centrioles in control cells and the satellite localization is lost in PCM1 KO cells. However, in control cells, Mib1 clearly localizes on satellites only and exclusively showed centriolar localization in PCM1 KO cells (Figure 4).

Figure 3, the failure to localize Cep90 to satellites could just be a consequence of its level in the cell. Figure 1 shows that it is strongly reduced in the absence of PCM1.

We have now examined Cep90 levels and show these data in new Figure 3—figure supplement 1. We found that reintroduction of the 1-1200 fragment could partially rescue the Cep90 protein level to a level similar to the full length PCM1, although satellite localization was not rescued (Figure 3), suggesting that the absence of Cep90 on satellites is not caused by reduced protein levels.

Reviewer #2:

*The function of centriolar satellites, which form a dispersed halo around the centriole, has been the topic of extensive speculation but has not been analyzed rigorously. This manuscript presents a solid set of data that demonstrate the importance of the pericentriolar satellite protein PCM1 in formation of cilia in RPE1 cells. Previous knockdown experiments had shown a mild effect of PCM1 deletion on ciliogenesis. Here, the investigators make null mutant cells using CRISPR/Cas and document the complete absence of cilia in mutant cells. They also show that PCM1 is required for the focal localization in satellites of all tested PCM proteins, demonstrating the central role of PCM1 in these structures. The investigators show that the phenotypes can be rescued by expression of full-length PCM1 and then use standard structure-function experiments to identify regions of PCM1 that are required to rescue the ciliogenesis defect and that bind to other satellite proteins. The authors then focus on Mib1, an E3 ubiquitin ligase, which was recently shown by another group to be enriched in centriolar satellites and suggested that Mib1 can suppress cilia formation. The authors show Mib1 and some other satellite proteins relocalize from satellites to centrioles in the PCM1 KO cells. They go on to show that siRNA knockdown of Mib1 in the PCM1 KO cells restores some ciliogenesis (to about half the level in control cells); this is an important and interesting result that shows that PCM1 promotes ciliogenesis, at least in part, by removing Mib1 from the centriole. It would be helpful to show some images of these cilia to demonstrate whether these cilia are normal in structure and length. In a supporting experiment (Figure 5), the authors show that overexpression of wild-type Mib1, but not a catalytically inactive form of Mib1, inhibits cilia formation; the text describing this experiment is confusing and should be rewritten.*

We thank the reviewer for these comments. We have included the requested figure showing cilia in PCM1-/- cells with Mib1 knockdown in Figure 5—figure supplement 1. In these rescued cells, cilia indeed appear to be normal as determined by staining for GT335, and their length also appears normal.

For Figure 5, we have modified the Discussion accordingly to remove the confusing description.

*The authors then investigate what substrate of Mib1 at the centriole could regulate cilia formation and show that several centriolar and satellite proteins can be polyubiquitylated by Mib1. They focus on Talpid3, which is puzzling, as they state that there is no direct interaction between Mib1 and Talpid3 and that the proteins normally reside in different compartments, and the* in vivo *interactions are therefore likely to be indirect, and the effects on Talpid3 levels could be mediated indirectly by other Mib1 targets.*

In studies that we did not present, we identified Mib1 after ectopic expression of Talpid3 and immuno-purification of associated polypeptides in mass spectrometric sequencing experiments. These results can be explained by the ability of Mib1 to interact with over-expressed Talpid3. It is possible that the impact on Talpid3 levels could be mediated indirectly by other Mib1 targets, and we have now included this possibility in our Discussion. However, several pieces of evidence support our model. First, ablation of PCM1 leads to the translocation of Mib1 from centriolar satellites to centrioles and decreased Talpid3 protein levels on centrioles (Figure 3, Figure 4, and 7 and Figure 2—figure supplement 1). Second, Mib1 interacts with Talpid3 and promotes poly-ubiquitylation of Talpid3 (Figure 6). Third, over-expression of catalytically active Mib1 (with or without an attached PACT domain) results in decreased Talpid3 on centrioles, whereas knock-down of Mib1 rescues Talpid3 levels on centrioles in PCM1-/- cells (Figure 7 and new Figure 8). We present the most straightforward model to account for all of these data, although other models are certainly possible, including the likely possibility that other yet-to-be-defined substrates at the centriole are important for the regulation of ciliogenesis, in addition to Talpid3.

*In a final set of experiments, the authors show that Rab8 fails to localize to the centriole in PCM1 mutant cells and that some Rab8 can accumulate at the centriole when Mib1 is overexpressed, consistent with the hypothesis that sequestration of Mib1 away from the centriole by PCM1 allows Rab8 to be recruited to the mother centriole in an early stage of ciliogenesis. This logic of a PCM1-Mib1-Talpid3-Rab8 pathway is reasonable, but its* in vivo *relevance not entirely compelling, as there is no* in vivo *evidence for the role of Mib1 in cilia formation (Mib1 null mice show phenotypes consistent with loss of Notch signaling) and Rab8a null mutants are viable, which should be acknowledged by the authors.*

In our paper, we show that Mib1 is a negative regulator of ciliogenesis based on data in RPE1 cells, consistent with a previous report showing that knock-down of Mib1 promotes ciliogenesis in human RPE1 cells (Villumsen et al. 2013). In published papers reporting the Mib1 null mice, however, ciliogenesis was not studied. Indeed, as correctly stated by the reviewer, the dominant phenotype in Mib1 knockout mouse reflects defects in Notch signaling. It is also possible that the Mib1 knock-out could exhibit compensation, since there is a second Mib gene, Mib2. It would be interesting to examine ciliation in Mib1-/- mice, but since this has not been examined, these earlier studies do not have a direct bearing on our work. Further, as mentioned by the reviewer, Rab8 knock-out animals are indeed viable, but the likely explanation is that there is compensation by another Rab family member, in particular, Rab10, suggesting functional redundancy of these proteins during ciliogenesis. We agree with the reviewer that these observations merit discussion, and they have incorporated into the text.

[Editors’ note: the author responses to the re-review follow.]

*A few minor revisions are necessary to address the published work cited by the reviewer. The reviewer lists three papers to cite and whose citation will require some modification to the text.*Reviewer #1:

*The authors have done an excellent job of responding to my criticisms and feel that the new experiments and rewording have addressed my concerns. However, the new discussion has some issues that need to be addressed. Double Mib1-/-, Mib2-/- mouse knockouts have been made. Mib2-/- does not have much of a phenotype while the Mib1-/-, Mib2-/- look like the Mib1-/- animals (PMID: 18043734). Since the loss of Mib2 does not alter the Mib1 phenotype, the compensation argument in the mouse is weak. In addition, for the authors to make this argument, they need to show that Mib2 is not normally expressed in their cells and that the ectopic expression of Mib2 rescues the loss of Mib1.*

We thank the reviewer for pointing this out. We note that Mib1-/-, Mib2-/- mouse knockouts have been made and look like the Mib1-/- animals, displaying a dominant phenotype that reflects defects in Notch signaling (PMID: 18043734). However, we note again that ciliogenesis was not studied. It is not known whether Mib1-/- or Mib1-/-/Mib2-/- MEFs have a higher ciliation rate compared to wild-type cells, but since this has not been examined, it is not accurate to state that the knock-out study disagrees with our work. In our paper, using RPE1 cells, we show that Mib1 is a negative regulator of ciliogenesis based on data, consistent with a previous report showing that knock-down of Mib1 promotes ciliogenesis in the same cell line (Villumsen et al. 2013).

Nevertheless, we have modified the Discussion accordingly, raising the idea of the importance of investigating ciliogenesis in *MIB1^-/-^/MIB2^-/-^* knockouts. The paper reporting MIB1-/-/MIB2-/- double knockouts has now been cited.

The Rab8 discussion is superficial as there are two Rab8 isoforms. The knockout of both has little effect in the mouse (PMID: 24213529). Rab10 may be compensating but I don't know of any work that shows that Rab10 comes in to fill the role of Rab8 when Rab8a/b are missing.

We inadvertently omitted the citation for the Rab8a/b double knockout paper from the reference list, although it was cited in the text, and have corrected this oversight. We agree that it is important investigate whether/how Rab10 compensates in Rab8a/b double knockouts. We have therefore modified the Discussion accordingly, mentioning the importance of investigating Rab10 function in the Rab8a/b double knockout.

The BBS4 discussion is not correct as mice and other organisms lacking this gene still form cilia (PMID: 15173597).

We disagree with the reviewer’s comment. In the mouse, the function of BBS4 in ciliogenesis is still a matter of debate. There is evidence supporting the requirement for BBS4 in ciliogenesis in a recent study (PMID: 23716571), in which kidney cells from Bbs4 knockout mice show a significantly reduced ciliation rate (23% Bbs4−/− versus 48% WT; P < 0.001) and decreased cilium length. Moreover, knock-down of BBS4 in zebrafish leads to defective ciliogenesis in Kupffer's vesicle (PMID: 16399798). These data, in aggregate, suggest that BBS4 plays an important role in ciliogenesis in animals. These papers, together with the earliest BBS4-/- paper (PMID: 15173597), were added to the references.